# The kinase PDK1 is critical for promoting T follicular helper cell differentiation

Zhen Sun[1†], Yingpeng Yao[1†], Menghao You[1†], Jingjing Liu[1], Wenhui Guo[1], Zhihong Qi[1], Zhao Wang[1], Fang Wang[1], Weiping Yuan[2], Shuyang Yu[1]*

[1]State Key Laboratory of Agrobiotechnology, College of Biological Sciences, China Agricultural University, Beijing, China; [2]State Key Laboratory of Experimental Hematology, Institute of Hematology and Blood Diseases Hospital, and Center for Stem Cell Medicine, Chinese Academy of Medical Sciences and Peking Union Medical College, Tianjin, China

**Abstract** The kinase PDK1 is a crucial regulator for immune cell development by connecting PI3K to downstream AKT signaling. However, the roles of PDK1 in CD4[+] T cell differentiation, especially in T follicular helper (Tfh) cell, remain obscure. Here we reported PDK1 intrinsically promotes the Tfh cell differentiation and germinal center responses upon acute infection by using conditional knockout mice. PDK1 deficiency in T cells caused severe defects in both early differentiation and late maintenance of Tfh cells. The expression of key Tfh regulators was remarkably downregulated in PDK1-deficient Tfh cells, including *Tcf7*, *Bcl6*, *Icos*, and *Cxcr5*. Mechanistically, ablation of PDK1 led to impaired phosphorylation of AKT and defective activation of mTORC1, resulting in substantially reduced expression of Hif1α and p-STAT3. Meanwhile, decreased p-AKT also suppresses mTORC2-associated GSK3β activity in PDK1-deficient Tfh cells. These integrated effects contributed to the dramatical reduced expression of TCF1 and ultimately impaired the Tfh cell differentiation.

*For correspondence:
ysy@cau.edu.cn

[†]These authors contributed equally to this work

**Competing interests:** The authors declare that no competing interests exist.

## Introduction

The serine/threonine kinase 3-phosphoinositide-dependent protein kinase 1 (PDK1) is a critical metabolic regulator connecting PI3K and downstream molecules (*Park et al., 2009*). PDK1 is crucial for multiple types of immune cell development, such as hematopoietic stem cells, B cells, NK cells, T cells, and cytolytic CD8[+] cells (*Park et al., 2009*; *Baracho et al., 2014*; *Venigalla et al., 2013*; *Yang et al., 2015*; *Finlay et al., 2012*). Despite these profound effects of PDK1 on the regulation of various immune cell subsets, its role in CD4[+] T helper cells, especially T follicular helper (Tfh) cell differentiation, has not been experimentally determined.

Upon foreign antigens challenge, naïve CD4[+] T cells differentiate into functionally distinct subsets. Tfh cells are a specialized population that provides cognate help in germinal center (GC) to facilitate immunoglobulin affinity maturation and heavy-chain class switching, and ultimately promote generation of high-affinity antibodies of diverse isotypes, long-lived plasma cells, and memory B cells (*Crotty, 2014*). The characteristic features of Tfh cells are defined as expression of the CXCR5, ICOS, PD-1, Bcl-6, and IL-21 (*Crotty, 2019*). Differentiation of Tfh cell relies on precisely orchestrated transcriptional program, which lies in the mutually antagonistic Bcl-6-Blimp1 as the core regulatory axis. Bcl-6 is indispensable for the divergence of CD4[+] T cells into Tfh lineage, whereas Blimp1 programs Th1 cell formation (*Johnston et al., 2009*; *Yu et al., 2009*; *Nurieva et al., 2009*). TCF1 (encoded by *Tcf7*) supports Tfh cell specification by promoting Bcl-6 and repressing Blimp1 expression, revealing a key role of TCF1 in the regulation of the Bcl-6-Blimp1 axis (*Shao et al., 2019*; *Wu et al., 2015*; *Choi et al., 2015*; *Xu et al., 2015*).

Compared with other T helper cells, the differentiation of Tfh is more dependent on signals provided by both TCR and co-receptors (*Deenick et al., 2010*; *Baumjohann et al., 2013*). Except sustained TCR signals, Tfh cells also express high levels of many co-receptors for their generation and function, including CD28 and ICOS (*Qin et al., 2018*). The PI3K signaling via AKT is responsible for transduction of Tfh cell-dependent TCR signals and ICOS co-stimulation and essential for Tfh cell differentiation. Previous studies have revealed inactivation of PI3K catalytic subunit p110δ or regulatory subunit p85α almost completely abolishes Tfh cell differentiation (*Rolf et al., 2010*; *Leavenworth et al., 2015*), whereas mice that express the E1020K activating mutant of p110δ or have a T cell-specific deficiency of PTEN exhibit enhanced Tfh cell formation (*Preite et al., 2018a*; *Shrestha et al., 2015*). Moreover, FoxO1, which is suppressed by AKT, restrains Tfh cell differentiation (*Stone et al., 2015*). Besides, PI3K-mediated signaling pathways involving mTORC1 and mTORC2 are also critical for Tfh cells. mTORC1 induces p-S6 and Glut1 expression to promote protein synthesis and cell proliferation, essential events for Tfh cell specification (*Zeng et al., 2016*). Unlike mTORC1, mTORC2 programs Tfh cell differentiation by decreasing FoxO1 activity and transcriptionally regulates signature programming of Tfh cells, including *Bcl6*, *Cxcr5*, and *Tcf7* (*Hao et al., 2018*). Correspondingly, Yang et al. reported mTORC2-deficient CD4$^+$ T cells show reduced p-GSK3β, β-catenin, and TCF1 level, establishing a positive link between PI3K/AKT and Wnt/β-catenin/TCF1 signaling (*Yang et al., 2016*). Besides, mTOR-dependent Hif activity is also crucial for Tfh cell development (*Cho et al., 2019*).

In this study, we explore the knowledge gap whether and how PDK1 plays essential roles in Tfh cell differentiation elicited by acute viral infection and protein immunization. Our data indicated that PDK1 is intrinsically required for Tfh cell formation and effector functions, which is important to understanding the nature of the Tfh cell development.

## Results

### PDK1 promotes Tfh cell differentiation and GC Bcell responses

To elucidate whether PDK1 regulates Tfh cell differentiation, we first evaluated the expression of PDK1 in bifurcation of effector CD4$^+$ T cells into Tfh or Th1 cells upon acute viral infection (*Xu et al., 2015*). We infected wild-type C57BL/6J mice with LCMV Armstrong strain and observed elevated expression level of PDK1 in Tfh cells compared with naïve CD4$^+$ T or Th1 cells on day 8 post-infection (8 dpi), which suggested potential roles of PDK1 in Tfh cells (*Figure 1A*). To further investigate the function of PDK1 in Tfh cells, we generated conditionally knockout mice (*Pdk1*$^{fl/fl}$::*Cd4*-Cre mice), and the deletion efficiency in CD4$^+$ T cells was further confirmed by quantitative RT-PCR (*Figure 1B*). Next, *Pdk1*$^{fl/fl}$::*Cd4*-Cre mice and their wild-type littermates (WT) were infected with LCMV Armstrong. On 8 dpi, generations of Tfh cells (26.5-fold) and Th1 cells (2.6-fold) were substantially decreased in *Pdk1*$^{fl/fl}$::*Cd4*-Cre mice (*Figure 1C,D*), indicating PDK1 has a more crucial role for Tfh cell differentiation. Moreover, GC Tfh cells were also significantly reduced in PDK1-deficient mice (*Figure 1C,D*). Consistently, the expression levels of PD-1, ICOS, and Bcl-6 were much lower in *Pdk1*$^{fl/fl}$::*Cd4*-Cre Tfh cells than those of WT cells (*Figure 1E,F*). Collectively, these data indicated that PDK1 is essential for Tfh cell differentiation during viral infection.

High expression of CXCR5, which is the B-cell homing chemokine CXCL13-responding receptor, allows Tfh cells to access the B-cell follicle (*Crotty, 2011*), Tfh cells then support GC formation and effective humoral immunity (*Crotty, 2019*). Thus, we first analyzed Tfh cell migratory response toward CXCL13. *Pdk1*$^{fl/fl}$::*Cd4*-Cre Tfh cells exhibited severe defects in migratory response toward CXCL13 (*Figure 1G*), indicating impairment of the follicular migratory potential of Tfh cells in the absence of PDK1 may constrain GC responses. We next examined the effects on GC B-cell responses due to PDK1 deficiency. Flow cytometry analysis revealed GC B cells and plasma cells were profoundly impaired in *Pdk1*$^{fl/fl}$::*Cd4*-Cre mice compared with WT mice (*Figure 1H,I*). In concert with decreased GC B cells, PNA$^+$ GCs areas in spleens of *Pdk1*$^{fl/fl}$::*Cd4*-Cre mice were also smaller than those in WT littermates (*Figure 1J*). Correspondingly, the concentrations of LCMV-specific IgG in the serum of *Pdk1*$^{fl/fl}$::*Cd4*-Cre mice were much lower than WT mice on 8 dpi and 56 dpi (*Figure 1K*). These data corroborated an indispensable role of PDK1 in Tfh and GC B-cell differentiation as well as antibody production.

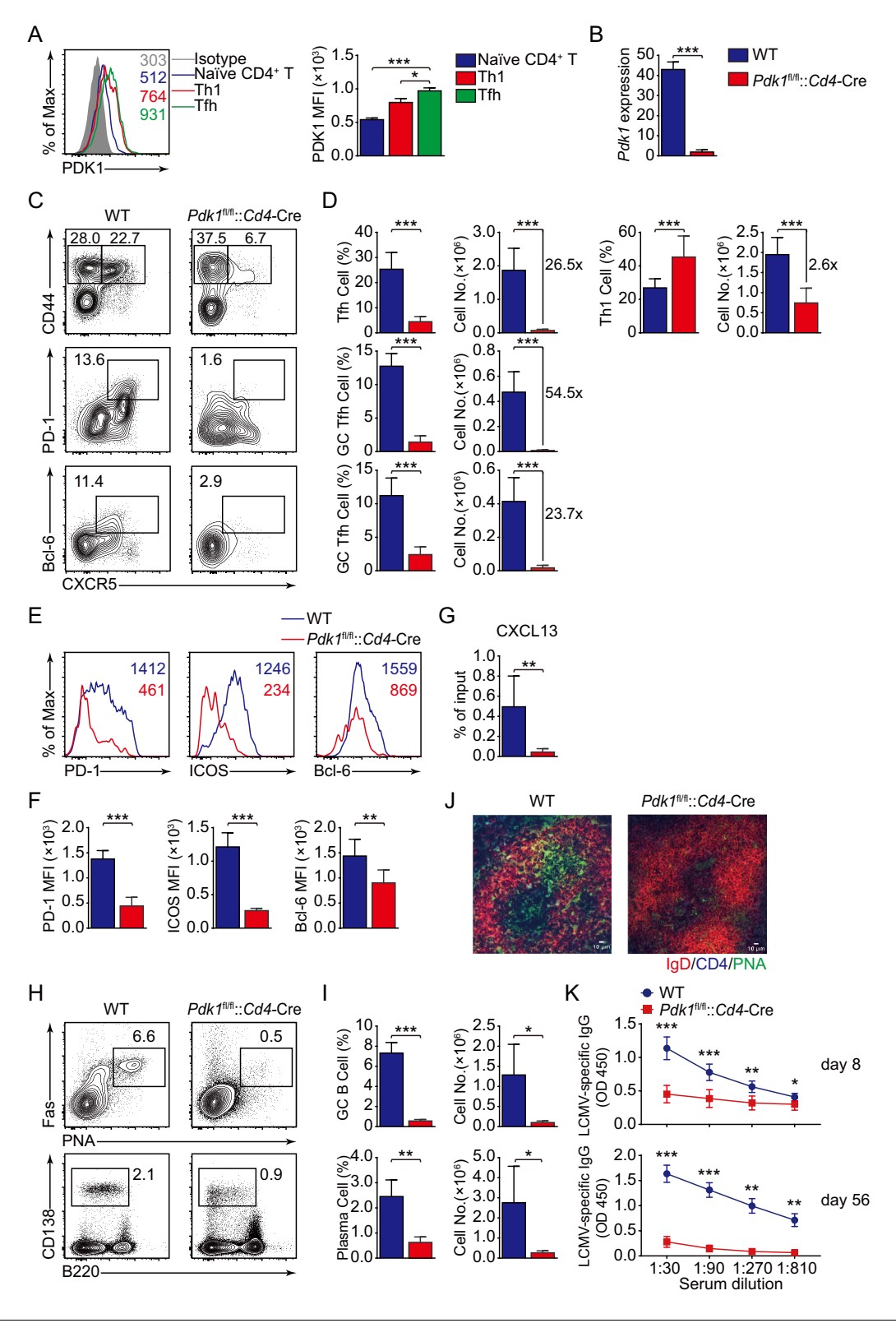

**Figure 1.** PDK1 supports Tfh cell differentiation and effector functions. (**A**) Flow cytometry analysis of PDK1 expression in naïve CD4$^+$ T (CD62L$^{hi}$CD44$^{lo}$), Th1, and Tfh cells from C57BL/6J mice on 8 dpi. Quantification of PDK1 MFI is shown on the right (n = 8). (**B**) Quantitative RT-PCR analysis of *Pdk1* abundance in naïve CD4$^+$ T cells, from WT and *Pdk1*$^{fl/fl}$::*Cd4*-Cre mice (*n* = 3). (**C, D**) Flow cytometry analysis of CD44$^+$CXCR5$^+$ Tfh cells and CD44$^+$CXCR5$^-$ Th1 cells gated on total CD4$^+$ T cells (top panel), or PD-1$^{hi}$CXCR5$^+$ GC Tfh cells (middle panel) and Bcl-6$^{hi}$CXCR5$^+$ GC Tfh cells

*Figure 1 continued on next page*

Figure 1 continued

(bottom panel) gated on CD44$^+$CD62L$^-$CD4$^+$ T cells from spleens of WT and *Pdk1*$^{fl/fl}$::*Cd4*-Cre mice on 8 dpi with representative contour plots and cumulative data in (C) and (D), respectively (n ≥ 6). (E, F) Expression of PD-1, ICOS, and Bcl-6 on Tfh cells (CD4$^+$CD44$^+$CXCR5$^+$) was analyzed by flow cytometry with representative histograms and quantification data in (E) and (F), respectively (n = 6). (G) Chemotaxis transwell assay for Tfh cells. Splenocytes from WT and *Pdk1*$^{fl/fl}$::*Cd4*-Cre mice on 8 dpi were added to a transwell plate and migration in the presence of CXCL13 was assessed (n ≥ 5). (H, I) Flow cytometry analysis of splenic PNA$^+$Fas$^+$ GC B cells (top panel) and B220$^-$CD138$^+$ plasma cells (bottom panel) from WT and *Pdk1*$^{fl/fl}$::*Cd4*-Cre mice on 8 dpi with representative contour plots and cumulative data in (H) and (I), respectively (n = 4). (J) Confocal microscopy analysis of GC histology in spleen sections from WT and *Pdk1*$^{fl/fl}$::*Cd4*-Cre mice on 8 dpi. Green: PNA, red: IgD, blue: CD4; scale bar: 10 μm. (K) LCMV-specific IgG concentration of sera from WT and *Pdk1*$^{fl/fl}$::*Cd4*-Cre mice on day 8 (top panel) and 56 (bottom panel) post-infection was measured by ELISA (n ≥ 7). Data are representative of at least two independent experiments (A, B, E–G, H–I, K) or pooled from three independent experiments (C, D). Error bars represent SD. *p<0.05, **p<0.01, and ***p<0.001 (Student's t-test).

The online version of this article includes the following source data for figure 1:

**Source data 1.** PDK1 supports Tfh cell differentiation and effector functions.

To validate the findings by using LCMV Armstrong infection approach, we also analyzed Tfh cell response upon KLH immunization. Consistent with the results shown above, loss of PDK1 led to significant reduction of Tfh cells as well as impaired expression of Tfh markers on day 8 post-KLH immunization (*Figure 2A–D*). Besides, both GC B cells and plasma cells were decreased in *Pdk1*$^{fl/fl}$::*Cd4*-Cre mice (*Figure 2E,F*). These data collectively demonstrated that PDK1 positively regulates Tfh cell differentiation and effector functions upon different antigens challenge.

In addition, we also analyzed the production of signature cytokines for other CD4$^+$ T cell subsets, such as IFNγ, IL-4, or IL-17a, in KLH-immunized WT and *Pdk1*$^{fl/fl}$::*Cd4*-Cre mice. CD4$^+$ T cells from *Pdk1*$^{fl/fl}$::*Cd4*-Cre mice showed greater expression of IFNγ, IL-4, and IL-17a than those in WT mice (*Figure 2—figure supplement 1*), which is consistent with previous report (*Yu et al., 2015*). These results indicated that PDK1 may also involve in other T helper cell differentiation under immunized condition.

## Intrinsic impact of PDK1 on Tfh cell development

To precisely clarify the cell-intrinsic role of PDK1 in regulating Tfh cell responses, we generated bone marrow (BM) chimeras by reconstituting lethally irradiated recipients (CD45.1$^+$ CD45.2$^+$) with a mixture of donor BM cells from *Pdk1*$^{fl/fl}$::*Cd4*-Cre or WT mice (CD45.2$^+$) with CD45.1$^+$CD45.2$^+$ WT mice (*Figure 3A*). After successful reconstitution (*Figure 3B*), we infected the chimeric mice with LCMV Armstrong and analyzed Tfh cells on 8 dpi. We first gated CD44$^+$CXCR5$^+$ Tfh cells, PD-1$^{hi}$CXCR5$^+$ GC Tfh cells, and Bcl-6$^{hi}$CXCR5$^+$ GC Tfh cells, and then the contributions of CD45.2$^+$ cells from WT or *Pdk1*$^{fl/fl}$::*Cd4*-Cre mice were analyzed. We found CD45.2$^+$ cells from *Pdk1*$^{fl/fl}$::*Cd4*-Cre mice only accounted for less than 1% of the total Tfh cells and GC Tfh cells, while CD45.2$^+$ cells of WT origin contributed to 23.1–29.7% among Tfh and GC Tfh cells (*Figure 3C–E*). Additionally, the expression levels of PD-1, ICOS, and Bcl-6 were substantially reduced in *Pdk1*$^{fl/fl}$::*Cd4*-Cre Tfh cells compared with WT cells (*Figure 3F,G*). Collectively, these data thus confirmed the intrinsic role of PDK1 in regulating Tfh cell differentiation.

## PDK1 is required for Tfh cells at both early and late stages

To investigate the role of PDK1 in early commitment or late maturation of Tfh cell differentiation, we bred *Pdk1*$^{fl/fl}$::*Rosa26*$^{CreER}$ mice with SMARTA mice to generate *Pdk1*$^{fl/fl}$::*Rosa26*$^{CreER}$::SMARTA mice, which enabled us to carry out the adoptive transfer assay by induced ablation of PDK1 with Tamoxifen at early stage of Tfh cell differentiation. We transferred WT or *Pdk1*$^{fl/fl}$::*Rosa26*$^{CreER}$::SMARTA cells into congenic recipient mice, which were then administrated with Tamoxifen followed by LCMV Armstrong infection (*Figure 4A,B*). On 3 dpi, both the frequency and cell numbers of Bcl-6$^+$CXCR5$^+$ Tfh cells of *Pdk1*$^{fl/fl}$::*Rosa26*$^{CreER}$::SMARTA mice were remarkably decreased compared with those of WT counterparts (*Figure 4C*). Moreover, both PDK1-deficient activated CD4$^+$ T cells and CXCR5$^+$ Tfh cells exhibited slower proliferation (*Figure 4D*), while the apoptosis was not altered in these cells (*Figure 4—figure supplement 1A*). We further sorted CXCR5$^+$ Tfh cells and performed quantitative RT-PCR analysis. The expression of Tfh cell-related genes *Tcf7*, *Cxcr5*, *Bcl6*, and *Pdcd1* was remarkably decreased in PDK1-deficient Tfh cells (*Figure 4E*). These results suggested that PDK1 is essential for proliferation and commitment of early Tfh cells.

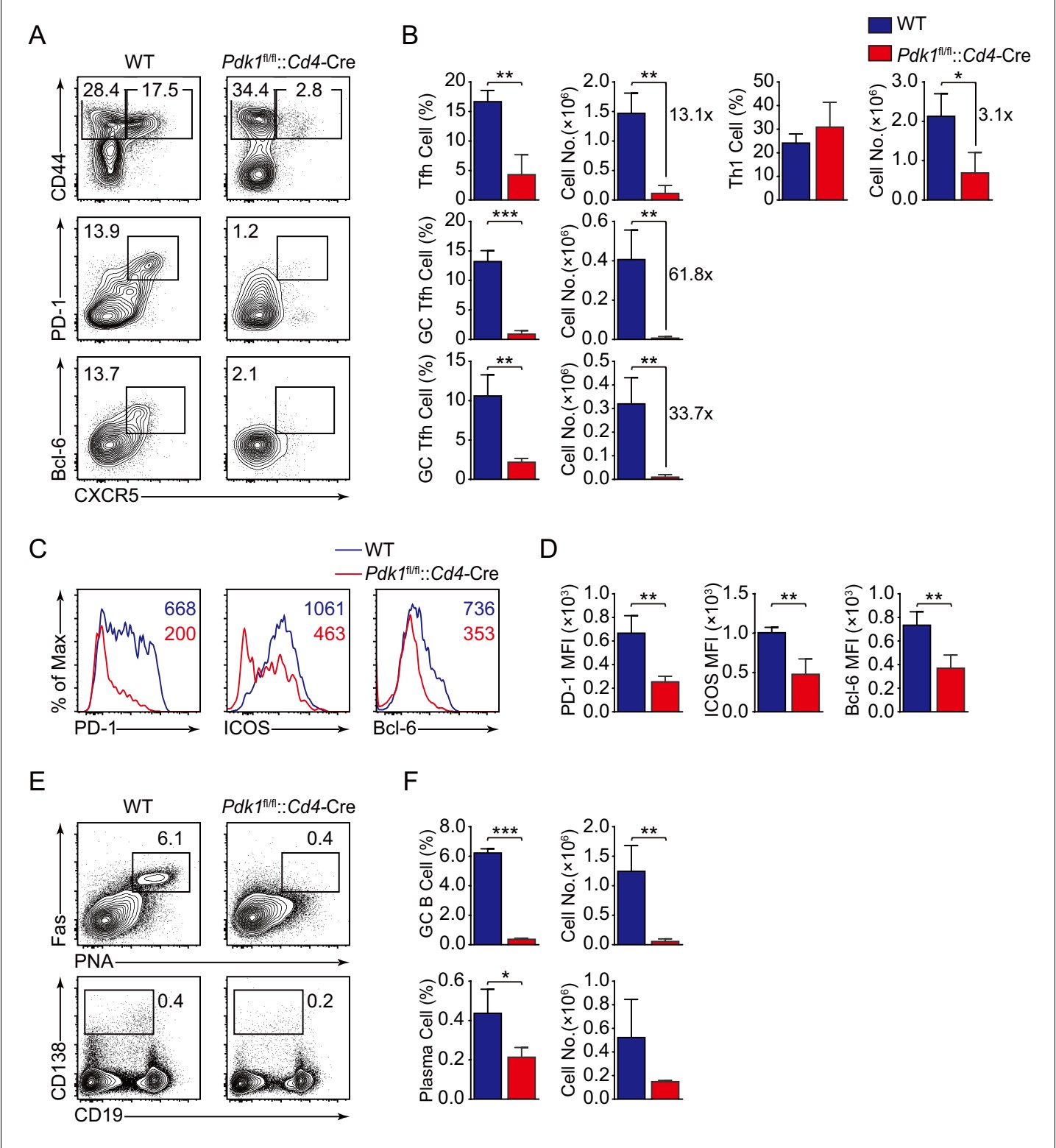

**Figure 2.** PDK1 is essential for Tfh cell differentiation upon protein immunization. (A, B) Flow cytometry analysis of CD44+CXCR5+ Tfh cells and CD44+CXCR5− Th1 cells gated on splenic CD4+ T cells (top panel) or PD-1hiCXCR5+ GC Tfh cells (middle panel) and Bcl-6hiCXCR5+ GC Tfh cells (bottom panel) gated on splenic CD44+CD62L−CD4+ T cells from WT and Pdk1fl/fl::Cd4-Cre mice on day 8 post-KLH immunization with representative contour plots and cumulative data in (A) and (B), respectively (n = 3). (C, D) Expression of PD-1, ICOS, and Bcl-6 on Tfh cells (CD4+CD44+CXCR5+) was analyzed by flow cytometry with representative histograms and quantification data in (C) and (D), respectively (n = 3). (E, F) Flow cytometry analysis of

*Figure 2 continued on next page*

Figure 2 continued

splenic PNA⁺Fas⁺ GC B cells (top panel) and B220⁻CD138⁺ plasma cells (bottom panel) from WT and *Pdk1*ᶠˡ/ᶠˡ::*Cd4*-Cre mice on day 8 post-KLH immunization with representative contour plots and cumulative data in (E) and (F), respectively (n = 3). Data are representative of two independent experiments. Error bars represent SD. *p<0.05, **p<0.01, and ***p<0.001 (Student's t-test).

The online version of this article includes the following source data and figure supplement(s) for figure 2:

**Source data 1.** PDK1 is essential for Tfh cell differentiation and GC responses upon KLH immunization.
**Figure supplement 1.** Analysis of T helper cells upon KLH immunization.
**Figure supplement 1—source data 1.** Analysis of distinct T helper subsets upon KLH immunization.

To specifically delete PDK1 at the late stage of Tfh cell differentiation, we treated WT or *Pdk1*ᶠˡ/ᶠˡ::*Rosa26*^CreER^ mice with tamoxifen from day 4 to day 7 post-viral infection (*Figure 4F,G*). On 8 dpi, we observed decreased Tfh cells and GC Tfh cells in tamoxifen-treated *Pdk1*ᶠˡ/ᶠˡ::*Rosa26*^CreER^ mice (*Figure 4H*). Moreover, both activated CD4⁺ T cells and CXCR5⁺CD44⁺ Tfh cells showed slower proliferation in the absence of PDK1 (*Figure 4I*). While the apoptosis of these two subsets was not affected (*Figure 4—figure supplement 1B*). These results indicated that PDK1 is also required for expansion and maintenance of late Tfh cell.

## PDK1-dependent Tfh cell transcriptomes

To further elucidate the mechanisms, we next explored how PDK1 deficiency impacts Tfh cell transcriptomes. CD44⁺SLAM^lo^ Tfh cells were sorted from LCMV Armstrong-infected *Pdk1*ᶠˡ/ᶠˡ::*Cd4*-Cre and WT mice on 8 dpi and subjected to RNA-seq. One thousand two hundred and thirty-eight upregulated and 354 downregulated genes in PDK1-deficient Tfh cells were identified by RNA-seq analysis (*Figure 5A*). Moreover, a dysregulation of diverse pathways was observed, in particular PI3K-AKT signaling pathway (*Figure 5B*). Then, we selected a Tfh gene set (*Choi et al., 2015*) for gene set enrichment analysis (GSEA). The Tfh signature genes were negatively enriched in *Pdk1*ᶠˡ/ᶠˡ::*Cd4*-Cre Tfh cells (*Figure 5C*). We further selected some interested differentially expressed genes (DEGs) from the RNA-seq results and confirmed their alterations by quantitative RT-PCR. We found the expression of Tfh cell-related genes *Cxcr5*, *Bcl6*, *Tcf7*, and *Icos* was decreased in PDK1-deficient Tfh cells compared with those of WT cells (*Figure 5D*). The expression of *Maf*, which encodes transcription factor c-Maf, inducing IL-21 expression in Tfh cells to support GC response (*Bauquet et al., 2009*), was decreased in PDK1-deficient Tfh cells compared with WT cells (*Figure 5D*). Moreover, the expression of *Hif1a*, which supports Tfh cell formation (*Cho et al., 2019*; *He et al., 2019*), was substantially lower in PDK1-null Tfh cells (*Figure 5D*). Whereas expression of other effector cells-relevant genes *Gzmb* (which encodes granzyme B), *Id2*, *Gata3*, and *Prdm1* was higher in PDK1-null Tfh cells than those in WT cells (*Figure 5D*). Taken together, these data strongly suggested that PDK1 is indispensable for maintaining Tfh cell identity.

Given Tfh cell differentiation is largely dependent on TCR and co-receptor pathways (*Fazilleau et al., 2009*; *Tubo et al., 2013*), we next questioned which signaling acts on upstream of PDK1 in Tfh cells. To achieve this goal, we performed GSEA with gene sets related to CD28- and ICOS-dependent pathways. *Pdk1*ᶠˡ/ᶠˡ::*Cd4*-Cre Tfh cells showed reduced expression of signatures in the gene set 'Up-regulated under anti-CD28' containing up-regulated genes upon anti-CD28 stimulation (*Figure 5E*). Moreover, *Pdk1*ᶠˡ/ᶠˡ::*Cd4*-Cre Tfh cells showed increased expression of signatures in the gene set 'Down-regulated under anti-ICOS-L' containing down-regulated genes upon ICOS-L blocking (*Figure 5E*). These analyses suggested that CD28- or ICOS-dependent PDK1 activity may involve in Tfh cell differentiation. To further validate this, we stimulated CD4⁺ T cells isolated from GP61-primed WT SMARTA mice with different stimuli combinations, including anti-CD3e plus anti-CD28, anti-CD3e plus anti-ICOS, anti-CD3e plus anti-CD28 plus anti-CD28, anti-ICOS only, or CD25 only (*Figure 5F,G*). We then measured the level of AKT phosphorylation at T308, which is an indicator of PDK1 activity. The results indicated that anti-ICOS only could elicit higher level of p-AKT^T308^, and combination of anti-ICOS plus anti-CD3e or/and anti-CD28 had a similar effect with anti-ICOS only (*Figure 5F,G*). These results further strengthen the notion that ICOS-dependent PDK1 activity is essential for Tfh cells, which is corresponding to the previous report that ICOS-driven PI3K signaling is indispensable for Tfh cell differentiation (*Gigoux et al., 2009*).

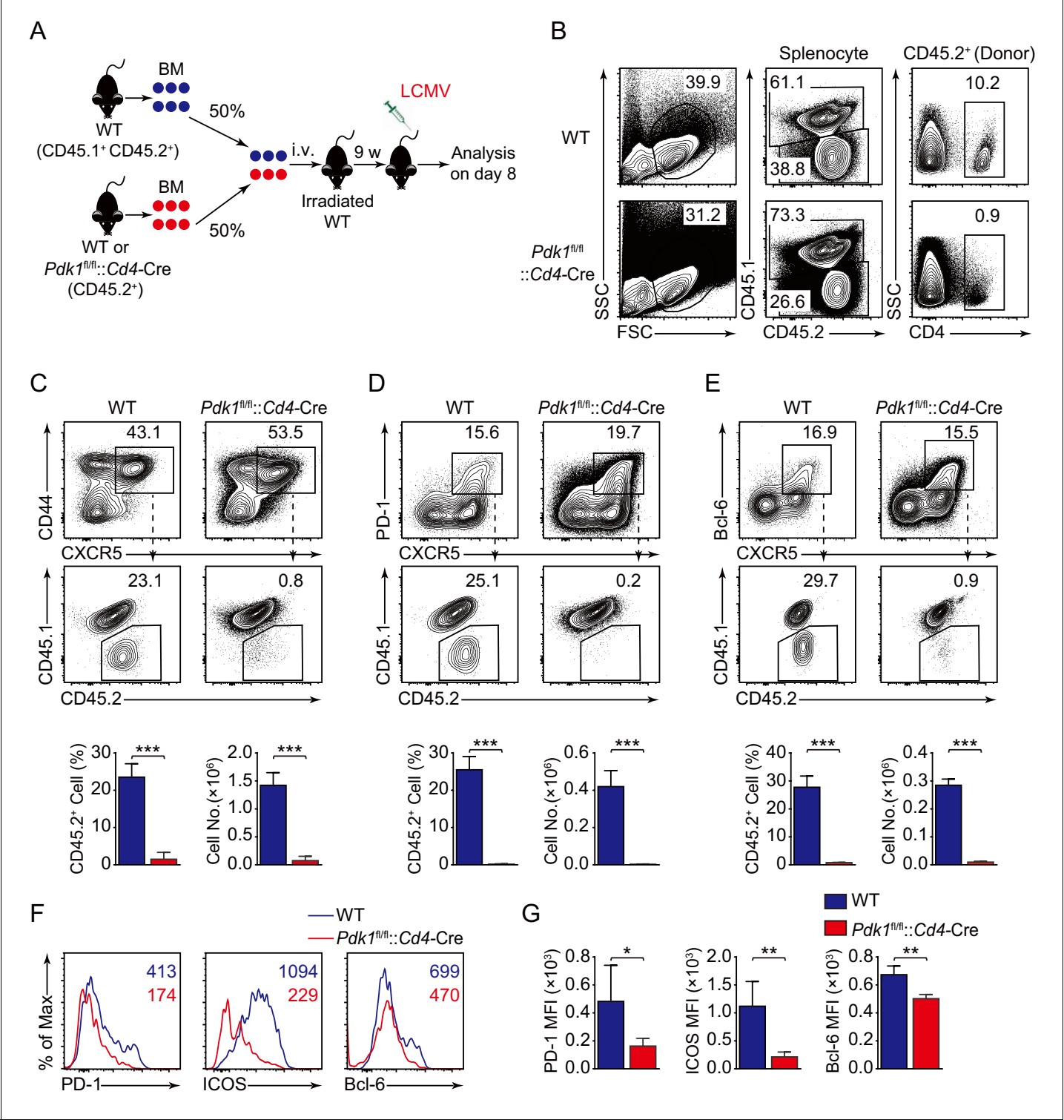

**Figure 3.** PDK1 intrinsically regulates Tfh cell differentiation. (**A**) Generation of bone marrow (BM) chimeric mice. BM cells from WT or _Pdk1_<sup>fl/fl</sup>::_Cd4_-Cre mice (CD45.2⁺) were mixed with WT (CD45.1⁺CD45.2⁺) competitor cells at a 1:1 ratio, and transferred to lethally irradiated WT recipients (CD45.1⁺CD45.2⁺). After 9 weeks reconstitution, the recipients were infected with LCMV and analyzed 8 days later. (**B**) Analysis of chimerism by flow cytometry. WT and _Pdk1_<sup>fl/fl</sup>::_Cd4_-Cre cells (CD45.2⁺) in CD4⁺ T cells of chimera mice were determined. (**C–E**) Flow cytometry analysis of competitive contributions by CD45.2⁺ cells to the total CD44⁺CXCR5⁺ Tfh (**C**), PD-1<sup>hi</sup>CXCR5⁺ GC Tfh (**D**), and Bcl-6<sup>hi</sup>CXCR5⁺ Tfh (**E**) cell population from recipients with representative contour plots and cumulative data (n = 4). (**F, G**) Detection of PD-1, ICOS, and Bcl-6 expression on CD44⁺CXCR5⁺ Tfh cells from

_Figure 3 continued on next page_

*Figure 3 continued*

recipients by flow cytometry with representative histograms and quantification data in (**F**) and (**G**), respectively (n = 4). Data are representative of two independent experiments. Error bars represent SD. *p<0.05, **p<0.01, and ***p<0.001 (Student's t-test).

The online version of this article includes the following source data for figure 3:

**Source data 1.** PDK1 intrinsically programs Tfh cell differentiation.

## PDK1 modulates TCF1 expression to program Tfh cell differentiation corresponding to both mTORC1 and mTORC2 signals

We next explored the potential targets of PDK1 involving in regulation on Tfh cells. We first looked at the effects of PDK1 deficiency on its downstream signaling by GSEA. We observed both Raptor and Rictor-activated genes were significantly enriched in WT Tfh cells, while Raptor and Rictor-suppressed genes were remarkably enriched in *Pdk1*^fl/fl^::*Cd4*-Cre Tfh cells. These results indicated that PDK1 and mTORC1/mTORC2 regulate a common subset of target genes in the Tfh cells (*Figure 6A*). We then determined these downstream signaling by flow cytometry analysis. We found *Pdk1*^fl/fl^::*Cd4*-Cre Tfh cells showed decreased amounts of both basal and phosphorylation of AKT at T308 and S473 compared with WT cells (*Figure 6B*), which were consistent with previous observations in B cells (*Venigalla et al., 2013*), suggesting phosphorylation of AKT at T308 and S473 was interdependent in Tfh cells. However, the phosphorylation level of PKCζ/λ, another target of PDK1, was comparable between PDK1-deficient and WT Tfh cells (*Figure 6—figure supplement 1A*). AKT phosphorylates FoxOs, and we observed both basal and phosphorylation levels of FoxO1/3a were substantially reduced in PDK1-deficient Tfh cells (*Figure 6C*). Based on these observations, we excluded FoxO1 as a potential downstream target due to its repression role in Tfh cell differentiation (*Stone et al., 2015*). AKT also activates mTORC1, a positive regulator for Tfh cell development (*Zeng et al., 2016*), and in line with the loss of AKT activation, mTORC1 activity was significantly decreased with the ablation of PDK1, as indicated by impaired S6 phosphorylation and reduced Hif1α expression (*Figure 6C*). Taken together, we concluded PDK1-dependent AKT activation is essential for Tfh cell commitment.

Previous studies have revealed AKT phosphorylates and inactivates GSK3β, which then negatively controls β-catenin/TCF1 axis (*Cross et al., 1995*; *Zhao et al., 2010*). Consistent with decreased p-AKT, phosphorylation of GSK3β was remarkably reduced in PDK1-deficient Tfh cells (*Figure 6C*). Moreover, expression level of TCF1 was much lower in PDK1-null Tfh cells than that of WT cells (*Figure 6C*). Decreased GSK3β phosphorylation in PDK1-deficient Tfh cells caused enhanced GSK3β activity (*Figure 6C*), which was accounted for impaired TCF1 level. It has recently reported that mTORC1-mediated STAT3 phosphorylation induced TCF1 expression in follicular regulatory helper (Tfr) cells (*Xu et al., 2017*). Similarly, we also observed decreased phosphorylation of STAT3 at Ser727 in PDK1-deficient Tfh cells (*Figure 6C*). These analyses collectively suggested that PDK1 may promote Tfh cell differentiation via both mTORC1- and mTORC2-dependent expression of TCF1. To validate this, WT or *Pdk1*^fl/fl^::*Rosa26*^CreER^::SMARTA cells transduced with TCF1 overexpressing retrovirus plasmid or empty vector (EV) were adoptively transferred into B6.SJL recipients, followed by Tamoxifen treatment and LCMV infection. On 8 dpi, EV-infected *Pdk1*^fl/fl^::*Rosa26*^CreER^::SMARTA CD4^+^ T cells remained defects in the generation of Tfh cells compared with WT cells, while TCF1 retrovirus promoted differentiation of *Pdk1*^fl/fl^::*Rosa26*^CreER^::SMARTA CD4^+^ T cells into Tfh cells (*Figure 6D*, *Figure 6—figure supplement 1B*). In addition, the cell numbers of *Pdk1*^fl/fl^::*Rosa26*^CreER^ Tfh cells were partially rectified by TCF1 (*Figure 6E*). Moreover, overexpression of a constitutive active form of STAT3 (STAT3-CA) could also rectify the defective Tfh cells in the absence of PDK1 (*Figure 6D,E*, *Figure 6—figure supplement 1B*). Besides, we also found forced expression of Bcl-6 or CXCR5 could partially restore the defective Tfh cell differentiation of PDK1-deficient CD4^+^ T cells (*Figure 6D,E*, *Figure 6—figure supplement 1B*). Collectively, our data demonstrated that TCF1 serves as one of critical regulators downstream of PDK1 in promoting Tfh cell differentiation.

## Discussion

The factors regulating Tfh cell differentiation, migration, and function are still being illustrated. Here, we focused on exploring the role of kinase PDK1 in the regulation of Tfh cell differentiation and

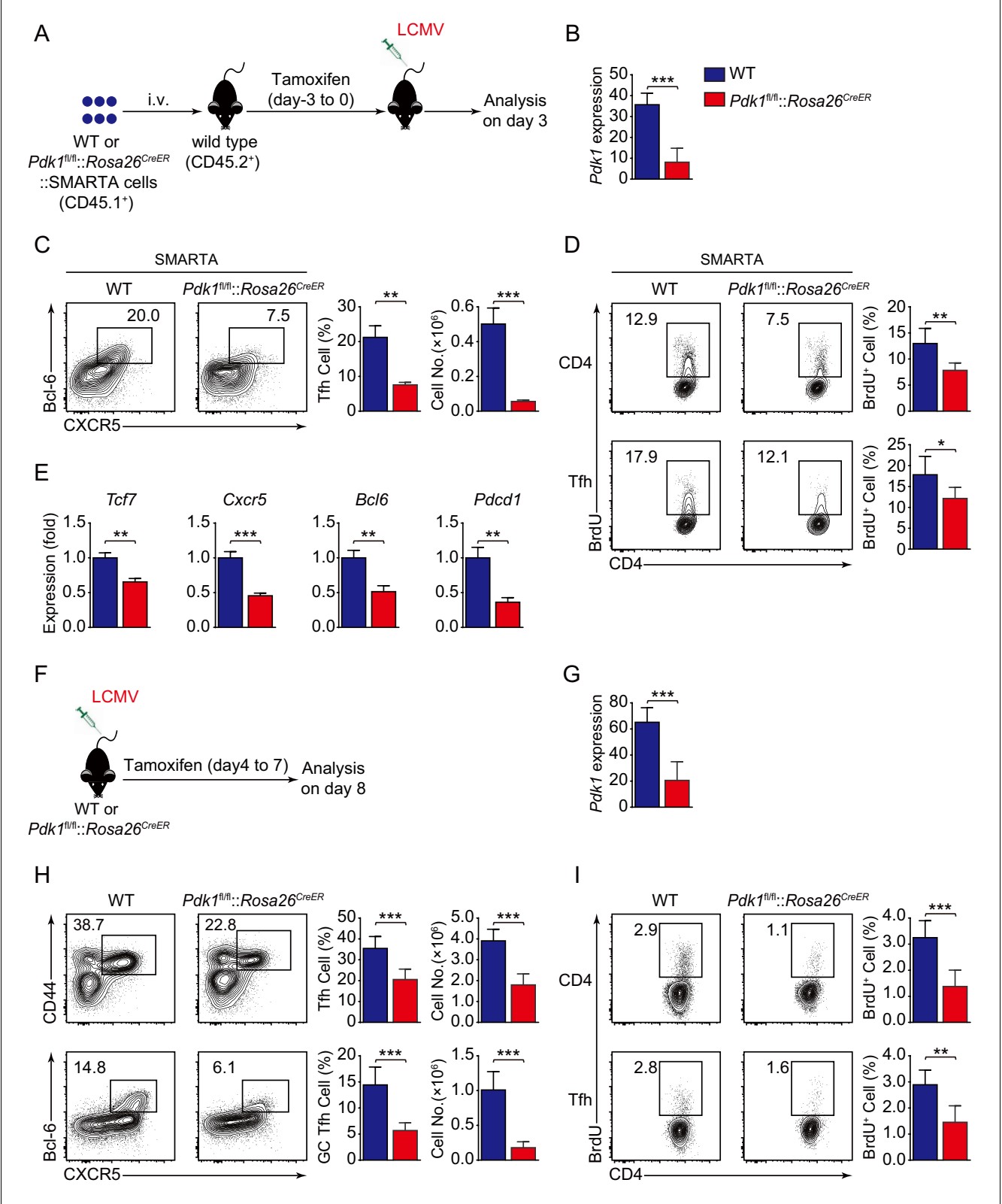

**Figure 4.** PDK1 is required for both early differentiation and late maintenance of Tfh cells. (**A**) Schematic of the SMARTA cell transfer system used for characterization of early Tfh cell commitment. SMARTA CD4⁺ T cells from *Pdk1*^fl/fl^::*Rosa26*^CreER^::SMARTA mice were transferred into C57BL/6J (CD45.2⁺) host mice, followed by Tamoxifen treatment for four consecutive days, LCMV infection, and analyzed on 3 dpi. (**B**) Quantitative RT-PCR analysis of *Pdk1* abundance in donor-derived CXCR5⁺ Tfh cells from recipients on 3 dpi as in (**A**) (n = 6). (**C**) Flow cytometry analysis of Bcl-6⁺CXCR5⁺

*Figure 4 continued on next page*

*Figure 4 continued*

Tfh cells gated on SMARTA CD4[+] T cells from recipients on 3 dpi with representative contour plots and cumulative data (n = 3). (D) Contour plots represents BrdU[+] cells gated on donor-derived activated CD4[+] T cells (top panel) and CXCR5[+] Tfh cells (bottom panel) from WT and *Pdk1*[fl/fl]::*Rosa26*[CreER]::SMARTA mice on 3 dpi. Cumulative data on frequency of BrdU[+] cells are shown on the right (*n* = 6). (E) Quantitative RT-PCR analysis of selected genes in donor-derived CXCR5[+] Tfh cells from recipients as in (A) (n = 3). (F) Schematic of the Tamoxifen-induced deletion system used for characterization of late Tfh cell differentiation. WT and *Pdk1*[fl/fl]::*Rosa26*[CreER] mice were treated with Tamoxifen from day 4 to day 7 post-LCMV infection and analyzed on 8 dpi. (G) Quantitative RT-PCR analysis of *Pdk1* abundance in Tfh cells from WT and *Pdk1*[fl/fl]::*Rosa26*[CreER] mice on 8 dpi as in (F) (n ≥ 5). (H) Flow cytometry analysis of CD44[+]CXCR5[+] Tfh cells (top panel) gated on CD4[+] T cells and Bcl-6[+]CXCR5[+] GC Tfh cells (bottom panel) gated on CD44[+]CD62L[-]CD4[+] T cells on 8 dpi with representative contour plots and cumulative data (n ≥ 5). (I) Contour plots represents BrdU[+] cells gated on activated CD4[+]CD44[+] T cells (top panel) and CD44[+]CXCR5[+] Tfh cells (bottom panel) from WT and *Pdk1*[fl/fl]::*Rosa26*[CreER] mice on 8 dpi. Cumulative data on frequency of BrdU[+] cells are shown on the right (n ≥ 5). Data are representative of at least three independent experiments (B–C, E, G–H) or pooled from two independent experiments (D). Error bars represent SD. *p<0.05, **p<0.01, and ***p<0.001 (Student's t-test).

The online version of this article includes the following source data and figure supplement(s) for figure 4:

**Source data 1.** PDK1 is essential for Tfh cell differentiation at both early and late stages.
**Figure supplement 1.** Detection of apoptosis of activated CD4[+] T cells and Tfh cells.
**Figure supplement 1—source data 1.** Analysis of apoptosis of CD4 T and Tfh cells.

effector function. The generation of Tfh cells was severely compromised in PDK1-deficient mice upon acute LCMV infection and KLH immunization. Correspondingly, the GC responses were also impaired as a consequence of defective Tfh cells. BM chimera results further revealed PDK1 controls Tfh cells in a cell-intrinsic fashion. By using different mice models, we validated that PDK1 is essential for both early expansion and late maintenance of Tfh cells. Taken together, these data support the notion that PDK1 is critical for the development and B-cell helper function of Tfh cells.

CD28 and ICOS are two key costimulatory receptors expressed by Tfh cells, both of which activate PI3Kδ and are essential for Tfh cell differentiation (*Preite et al., 2018b*). Previous study indicated that ICOS-dependent PI3K signal exerts nonredundant function in the generation of Tfh cells (*Gigoux et al., 2009*). By using GSEA and flow cytometry assay, we observed that ICOS functioned as upstream activator of PDK1 in CD4[+] T cells. Moreover, we found that Tfh cells exhibited higher expression of PDK1 than Th1 cells and naïve CD4[+] T cells. These results collectively suggested that ICOS-dependent PDK1 activity is pivotal for Tfh cells.

The PI3K-AKT signaling pathway is activated by various cell-surface receptors that are crucial for Tfh cell differentiation and function. It is well elaborated that PI3K is a critical component of pathway driving Tfh cell differentiation and GC formation supported by data from PI3K-targeting mice as well as mice and humans expressing activating mutants (*Rolf et al., 2010*; *Preite et al., 2018a*; *Preite et al., 2018b*; *Lucas et al., 2014*). Activation of PI3K facilitates the recruitment of PDK1 and AKT to the plasma membrane through their PH domains, which enables phosphorylation of T308 within the catalytic domain of AKT. To be fully activated, a second crucial residue (S473) located in a hydrophobic motif within AKT's regulatory domain must be phosphorylated by protein kinases like mTORC2 (*Fayard et al., 2010*). mTORC2-deficient mice exhibited severely impaired Tfh cells by activating Tfh cell repressor FoxO1 expression (*Zeng et al., 2016*; *Hao et al., 2018*), and we observed the phenotypes of *Pdk1*[fl/fl]::*Cd4*-Cre cells almost recapitulated the defects of mTORC2-deficient Tfh cells. mTORC2-deficient T cells exhibited decreased phosphorylation of p-AKT at Ser473 (*Zeng et al., 2016*), whereas, in our experimental system, we found a reduction of both phosphorylated T308 and S473 levels in PDK1-deficient Tfh cells, similar phenomena were also observed in PDK1-null B cells (*Venigalla et al., 2013*). Meanwhile, we observed a reduction of phosphorylated FoxO1/3a and basal FoxO1 levels, which is a downstream molecule of AKT. These results suggested that PDK1-AKT-FoxOs signaling may not be responsible for defective Tfh cells in PDK1-null mice, as FoxO1 is a negative regulator for Tfh cell formation (*Stone et al., 2015*). In addition, the level of phosphorylated S6 was severely decreased in PDK1-deficient Tfh cells, indicating impaired mTORC1 pathway in the absence of PDK1. Compromised mTORC1 activity in turn attenuates protein synthesis and cell proliferation, which are essential for Tfh cell differentiation (*Zeng et al., 2016*). Meanwhile, we found the expression of both *Hif1a* mRNA and Hif1α protein was significantly decreased in PDK1-deficient Tfh cells, which may also contribute to the defective phenotype as loss of Hif1α in CD4[+] T cells impairs Tfh cell differentiation (*Cho et al., 2019*;

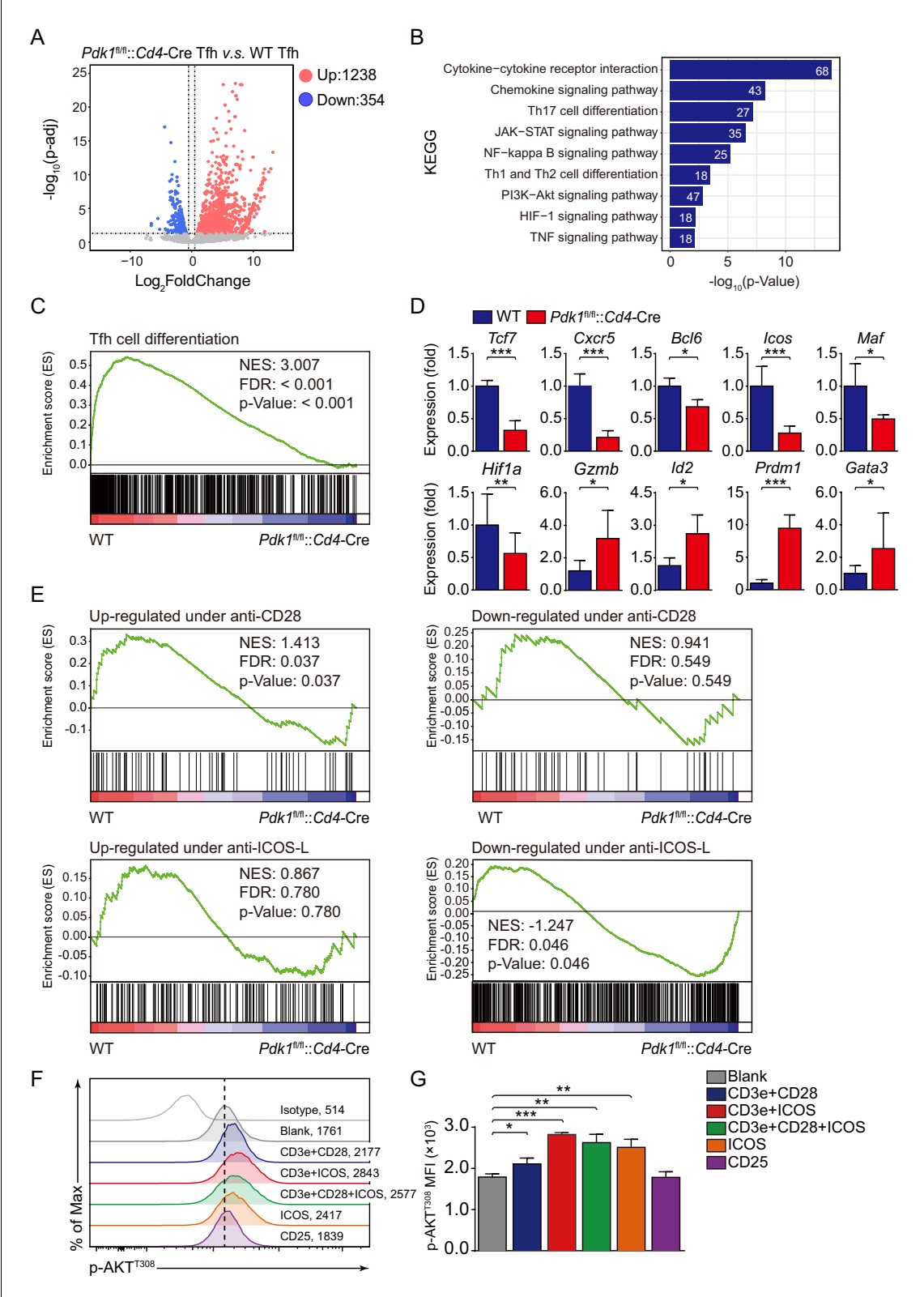

**Figure 5.** ICOS-dependent PDK1 promotes transcriptional program for Tfh cells. (**A**) RNA-seq analysis of *Pdk1*<sup>fl/fl</sup>::*Cd4*-Cre or WT Tfh cells sort-purified on 8 dpi. Volcano plot shows genes upregulated (red) or downregulated (blue) in *Pdk1*<sup>fl/fl</sup>::*Cd4*-Cre Tfh cells compared with WT cells. (**B**) KEGG pathway analysis of differentially expressed genes in *Pdk1*<sup>fl/fl</sup>::*Cd4*-Cre Tfh cells relative to their expression in WT Tfh cells. (**C**) GSEA of the Tfh cell gene signature in *Pdk1*<sup>fl/fl</sup>::*Cd4*-Cre Tfh cells relative to their expression in WT Tfh cells. (**D**) Quantitative RT-PCR analysis of selected genes in *Pdk1*<sup>fl/fl</sup>::*Cd4*-

*Figure 5 continued on next page*

*Figure 5 continued*

Cre and WT Tfh cells. Relative expression was normalized to WT cells (n ≥ 4). (E) GSEA of 'Up-regulated under anti-CD28', 'Down-regulated under anti-CD28', 'Up-regulated under anti-ICOS-L', and 'Down-regulated under anti-ICOS-L' gene sets in WT and *Pdk1*^fl/fl^::*Cd4*-Cre Tfh cells. (F, G) Flow cytometry analysis of p-AKT$^{T308}$ level on CD4$^+$ T cells from WT SMARTA cells, cultured in medium without any stimulus (blank), or stimulated with anti-CD3e + anti-CD28, anti-CD3e + anti-ICOS, anti-CD3e + anti-CD28 + anti-ICOS, anti-ICOS, and anti-CD25. Representative histogram plot and cumulative data are shown in (F) and (G), respectively (n = 3). Data are representative of at least two independent experiments (D, F, G). Error bars represent SD. *p<0.05, and ***p<0.001 (Student's t-test).

The online version of this article includes the following source data for figure 5:

**Source data 1.** ICOS-dependent PDK1 activity regulates Tfh cell transcriptional files.

*He et al., 2019*). These results collectively indicated that PDK1-dependent downstream molecular pathways are indispensable for Tfh cell biology.

Transcription factor TCF1 is expressed at an extremely high level in Tfh cells post-viral infection and exerts crucial roles in generation, maintenance, and effector functions of Tfh cells by repressing *Prdm1* and *Il2ra* and promoting *Bcl6* expression (*Shao et al., 2019*; *Wu et al., 2015*; *Choi et al., 2015*; *Xu et al., 2015*). It has been proposed that β-catenin, a coactivator of TCF1 that is negatively regulated by p-GSK3β, linking PI3K-AKT and TCF1 (*Yang et al., 2016*; *Zhou et al., 2010*). Besides, it has been reported that mTORC1 controls TCF1 expression in Tfr cells by phosphorylated STAT3 (*Xu et al., 2017* ). Correspondingly, we found both the p-GSK3β$^{S9}$ and p-STAT3$^{S727}$ levels were significantly decreased in PDK1-deficient Tfh cells. Furthermore, PDK1-null Tfh cells showed both impaired *Tcf7* mRNA and TCF1 protein level. Reconstituting PDK1-deficient CD4$^+$ T cells with TCF1 rectifies their defects in the generation of Tfh cells. Moreover, forced expression of STAT3-CA (constitutive active form of STAT3) could restore the Tfh cell numbers of PDK1-deficient mice. These observations collectively suggested that PDK1-dependent TCF1 expression contributes to Tfh cell formation.

In summary, the role of PDK1 in Tfh cell differentiation was extensively investigated in the current study. Our data further suggested PDK1 activates AKT in Tfh cells via phosphorylating both Thr308 and Ser473 residues. On the one side, p-AKT in turn activates mTORC1, and mTORC1 subsequently drives protein synthesis and Hif1α expression and supports TCF1 expression by p-STAT3 to sustain Tfh cell differentiation. On the other side, p-AKT also suppresses GSK3β activity and ultimately promotes TCF1 expression (*Figure 7*). Therefore, our study uncovers PDK1 as a critical regulator for Tfh cell development for both early expansion and late maintenance by mainly modulating TCF1 expression.

## Materials and methods

### Mice

*Pdk1*^fl/fl^ mice have been described before (*Lawlor et al., 2002*). *Cd4*-Cre, *Rosa26*^CreER^, and C57BL/6J (CD45.1 and CD45.2) mice were purchased from the Jackson Laboratory. SMARTA mice (specific for LCMV glycoprotein amino acids 66–77 presented by I-A$^b$) have been described (*Oxenius et al., 1998*) and were crossed to *Pdk1*^fl/fl^::*Rosa26*^CreER^ to generate *Pdk1*^fl/fl^::*Rosa26*^CreER^::SMARTA mice. All animals were on a C57BL/6J genetic background and used at 6–12 weeks of age. All animal experiments were performed according to protocols approved by the Institutional Animal Care and Use Committee at China Agricultural University.

### LCMV infection and KLH immunization

For acute viral infection model, mice were infected intraperitoneally (i.p.) with $2 \times 10^5$ pfu (plaque-forming units) of LCMV Armstrong strain in plain DMEM. For KLH immunization model, mice were immunized i.p. with 200 μg of KLH (1 mg/ml, Sigma–Aldrich) emulsified in CFA (1 mg/ml, Sigma–Aldrich). Eight days post-infection or immunization, the mice were sacrificed and splenocytes were examined.

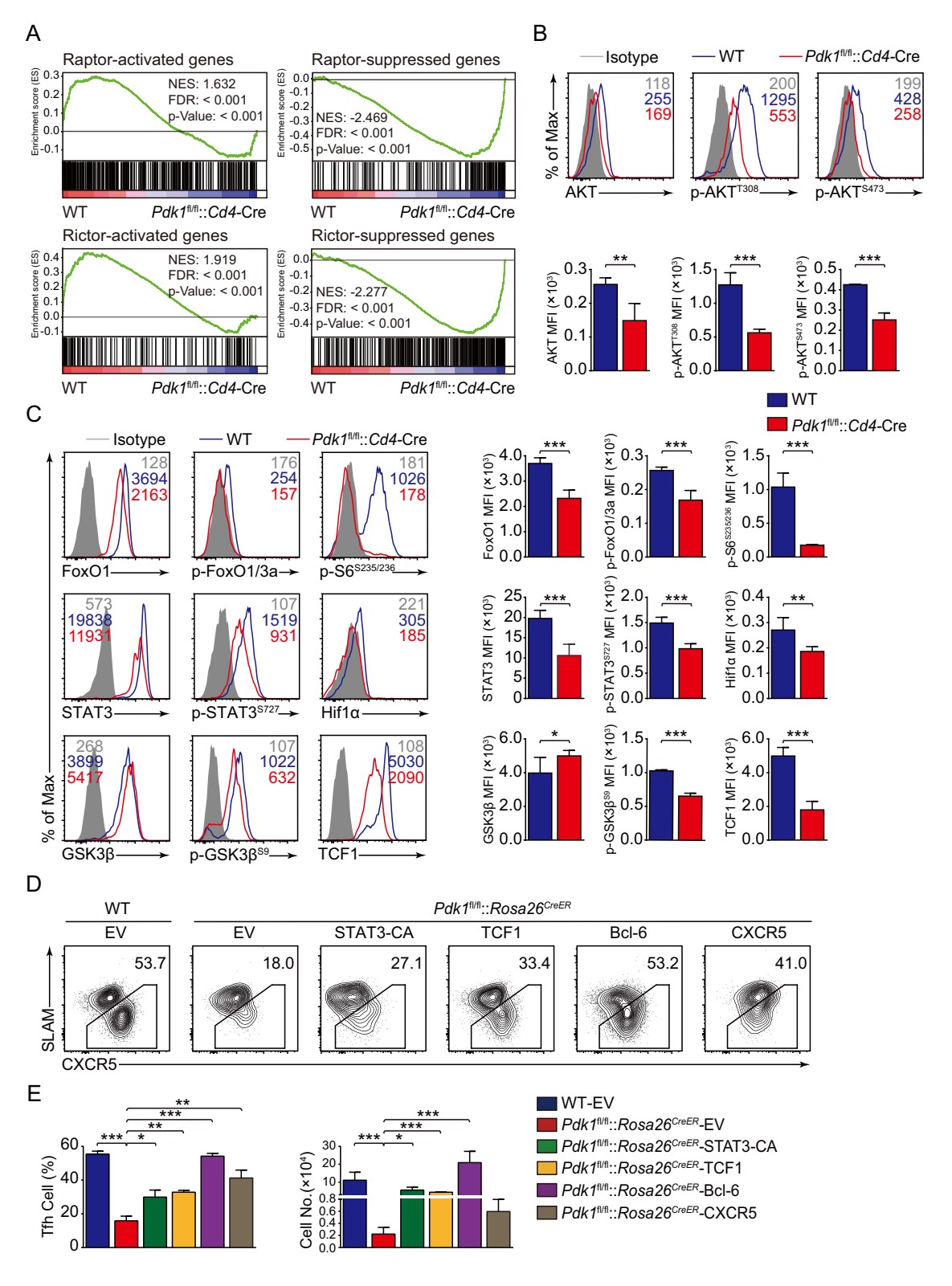

**Figure 6.** PDK1 deficiency impaired Tfh cell differentiation via mTORC1 and mTORC2 signal-dependent TCF1 expression. (A) GSEA of 'Raptor-activated genes', 'Raptor-suppressed genes', 'Rictor-activated genes', and 'Rictor-suppressed genes' gene sets in WT and $Pdk1^{fl/fl}::Cd4$-Cre Tfh cells. (B) Flow cytometry analysis of AKT, p-AKT$^{T308}$, and p-AKT$^{S473}$ levels on $Pdk1^{fl/fl}::Cd4$-Cre and WT Tfh cells on 8 dpi by flow cytometry with representative histograms and quantification data (n = 4). (C) Flow cytometry analysis of FoxO1, p-FoxO1/3a, STAT3, p-STAT3$^{S727}$, Hif1α, p-S6, GSK3β, *Figure 6 continued on next page*

*Figure 6 continued*

p-GSK3β$^{S9}$, and TCF1 levels on *Pdk1*$^{fl/fl}$::*Cd4*-Cre and WT Tfh cells on 8 dpi by flow cytometry with representative histograms and quantification data (n ≥ 4). (D, E) Flow cytometry analysis of Tfh populations from recipients adoptively transferred with STAT3-CA, TCF1, Bcl-6, or CXCR5 retrovirus-infected SMARTA cells on 8 dpi by flow cytometry with representative contour plots and cumulative data in (D) and (E), respectively (n ≥ 3). Data are representative of at least two independent experiments. Error bars represent SD. *p<0.05, **p<0.01, and ***p<0.001 (Student's t-test).

The online version of this article includes the following source data and figure supplement(s) for figure 6:

**Source data 1.** PDK1 regulates Tfh cell differentiation via mTORC1 and mTORC2 signal-dependent TCF1 expression.

**Figure supplement 1.** Analysis of p-PKCζ/λ level in Tfh cells and detection of overexpression level of relative genes in primed CD4$^+$ cells via retrovirus transduction.

**Figure supplement 1—source data 1.** Analysis of p-PKCζ/λ level and validation of overexpression efficiency.

## Flow cytometry

Single-cell suspensions were prepared from the spleens or peripheral blood for surface or intracellular staining. The fluorochrome-conjugated antibodies were used as follows: anti-CD4 (RM4-5), anti-CD25 (PC61), anti-CD44 (IM7), anti-CD45.1 (A20), anti-CD45.2 (104), anti-CD62L (MEL-14), anti-CD45R (RA3-6B2), anti-PD-1 (J43), anti-TCR Vα2 (B20.1), anti-IFNγ (XMG1.2), anti-IL-4 (11B11), anti-Foxp3 (FJK-16S) (from Thermo Fisher Scientific), anti-CD138 (281-2), anti-Fas (Jo2), anti-Bcl-6 (K112-91), anti-IL-17a (TC11-18H10) (from BD Biosciences), anti-SLAM (TC15-12F12.2), anti-ICOS (C398.4A)

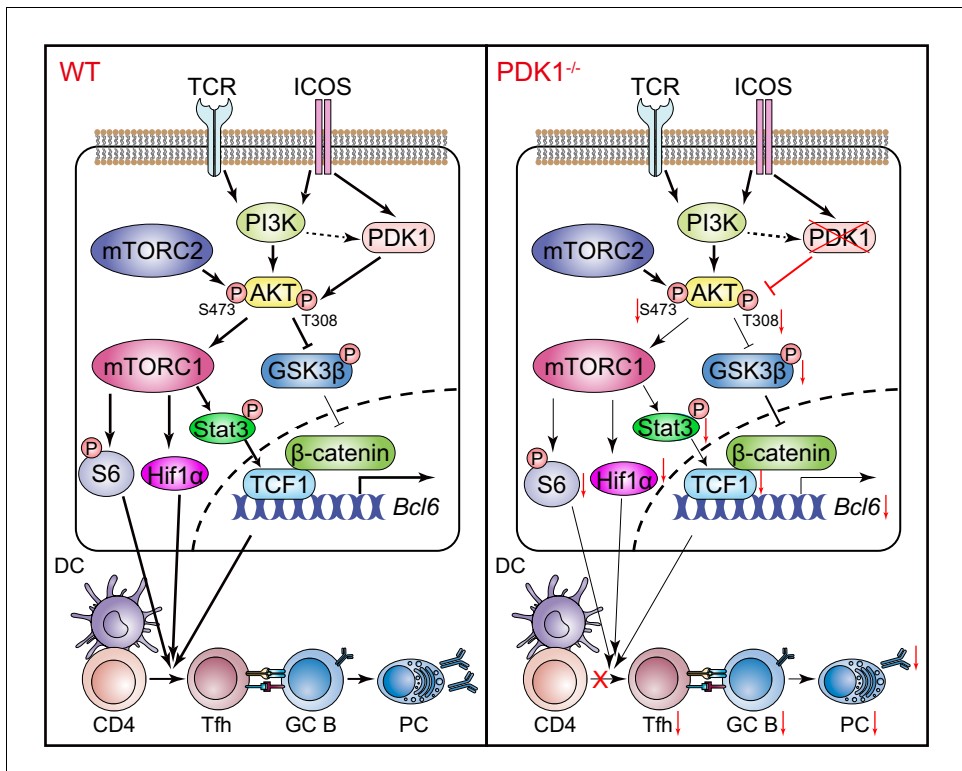

**Figure 7.** Working model of PDK1 in regulating Tfh cell differentiation. Left panel: In PDK1-sufficient cells, AKT gets activated by phosphorylation at Thr308 and Ser473. p-AKT activates mTORC1, and mTORC1 further phosphorylates S6 and supports Hif1α expression, promoting protein synthesis, proliferation, and metabolization. mTORC1 also phosphorates STAT3 to induce TCF1 expression. In addition, p-AKT also guards TCF1 activity through the inactivation of GSK3β, an inhibitor of TCF1 and β-catenin. Enhanced TCF1 contributes to Tfh cell differentiation, GC responses, and humoral immunity. Right panel: In PDK1-deficient cells, AKT remains inactivated by loss of phosphorylation at Thr308 and Ser473, which contributes to impaired mTORC1 activity and activities of downstream molecules, inducing p-STAT3-dependent TCF1 expression. In addition, inactivation of p-AKT leads to compromised p-GSK3β, resulting in increased GSK3β activity and subsequently inhibition on TCF1 level. Decreased TCF1 leads to impaired Tfh cell differentiation, GC responses, and humoral immunity.

(from BioLegend); PNA (Cat # FL-1071) (from Vector Laboratories), anti-TCF1 (C63D9), anti-PDK1 (Cat # 3062), anti-FoxO1 (C29H4), anti-Hif1α (D1S7W), anti-p-AKT$^{T308}$ (D25E6), anti-p-AKT$^{S473}$ (D9E), anti-p-STAT3$^{S727}$ (Cat # 9134), anti-p-GSK3β$^{S9}$ (D85E12), anti-p-S6$^{S235/236}$ (D57.2.2E), anti-p-FoxO1/3a (Cat # 9464), anti-p-PKCζ/λ (Cat # 9378) (from Cell Signaling Technology), and anti-AKT (Cat # AA326) and anti-GSK3β (Cat # AF1543) (from Beyotime Biotechnology). For detection of CXCR5, a three-step staining protocol was used with unconjugated anti-CXCR5 (2G8; BD Biosciences) as described before (*Yao et al., 2018*). For detection of Bcl-6, TCF1, and Foxp3, surface-stained cells were fixed and permeabilized with the Foxp3 Transcription Factor Staining Buffer Set (Thermo Fisher Scientific), followed by incubation with corresponding fluorochrome-conjugated antibodies. For the analysis of cytokine production, splenocytes well stimulated in vitro with PMA (Sigma–Aldrich) and Ionomycin (Sigma–Aldrich) at 37°C for 5 hr, then the cells were fixed and permeabilized with Cytofix/Cytoperm Fixation/Permeabilization Solution Kit (554714, BD Biosciences). For detection of phosphorylated proteins, stimulated cells were immediately fixed with Phosflow Lyse/Fix buffer (558049, BD Biosciences), followed by permeabilization with Phosflow Perm buffer I (557885, Biosciences), incubation with corresponding primary unconjugated antibodies and sequential staining with fluorochrome-conjugated donkey anti-rabbit IgG (poly4064; BioLegend). Flow cytometry data were collected on an LSRFortessa or a FACSVerse (BD Biosciences) and were analyzed with FlowJo software (TreeStar). The surface-stained cells were sort-purified on a FACSAria II (BD Biosciences).

## Adoptive transfer

For the adoptive transfer experiments, a total of $5 \times 10^6$ (analysis on day 3) SMARTA CD4$^+$ T cells from WT or *Pdk1*$^{fl/fl}$::*Rosa26*$^{CreER}$::SMARTA mice were adoptively transferred to congenic recipient mice. The recipient mice then received daily oval gavage with 2 mg of tamoxifen (Sigma–Aldrich) diluted in corn oil for 4 days, followed by intravenously (i.v.) infection with $2 \times 10^6$ pfu of LCMV Armstrong. On 8 dpi, the recipient mice were sacrificed, and the splenocytes were analyzed by flow cytometry.

Retroviral transduction pMIG-*Tcf7*, pMIG-*Bcl6*, pMIG-*Cxcr5*, MIG-*Stat3-CA*, and pMIG-R1 retroviral vectors were used to produce retrovirus from the HEK293 T cell lines. *Pdk1*$^{fl/fl}$::*Rosa26*$^{CreER}$ or WT SMARTA mice were i.v. injected with 200 µg of GP61 peptide (GLKGPDIYKGVYQFKSVEFD) to prime the SMARTA CD4$^+$ T cells. Sixteen hours later, the splenic CD4$^+$ T cells were isolated and infected by spinofection at 2100 rpm, 32°C for 90 min, and then cultured overnight in the presence of mIL-2 (20 ng/ml, PEPROTECH) and GP61 peptide (250 nM). The spinofection was repeated next day, and a total of 0.5 to $1 \times 10^6$ retrovirally infected SMARTA CD4$^+$ T cells (CD45.2$^+$) were i.v. transferred into CD45.1$^+$ recipients. The recipient mice received daily oval gavage with 2 mg of tamoxifen (Sigma–Aldrich) diluted in corn oil for 3 days, followed by i.p. infected with $2 \times 10^5$ pfu of LCMV Armstrong. The recipient mice were sacrificed, and spleens were examined 8 dpi.

## BM chimeras

BM chimeras were generated as previously described (*Liu et al., 2019*). Briefly, $2 \times 10^6$ BM cells of a 1:1 mixture from *Pdk1*$^{fl/fl}$::*Cd4*-Cre or WT donor mice (CD45.2$^+$) and CD45.1$^+$CD45.2$^+$ competitor mice were i.v. transferred into lethally irradiated (7.5 Gray each) wild-type CD45.1$^+$CD45.2$^+$ recipients. After 9 weeks reconstitution, recipient mice were infected with LCMV Armstrong. The recipient mice were sacrificed and spleens were examined 8 dpi.

## Enzyme-linked immunosorbent assay (ELISA)

ELISA to detect LCMV-specific IgG was performed as previously described (*Yao et al., 2018*). Briefly, Nunc MaxiSorp flat-bottom 96-well microplates (Thermo Fisher Scientific) were coated with LCMV-BHK21 lysates overnight. Plates were blocked with phosphate-buffered saline (PBS) + 0.05% Tween-20 + 1% BSA (PBST-B) for 30 min at room temperature. After washing, serum samples were added in serial dilution by PBST and incubated for 60 min at room temperature followed washing by PBST. Next, horseradish peroxidase-conjugated goat-anti-mouse IgG secondary antibody (Bethyl Laboratories) was added at 1:5000 in PBST-B for 60 min at room temperature. The LCMV-specific IgG was detected by coupling with TMB substrate (BioLegend). The absorbance at 450 nm was read on a microplate reader (TECAN).

## Immunofluorescence staining

Spleens from mice infected with LCMV Armstrong were snap-frozen in OCT medium (Sakura Fine-tek) by liquid nitrogen. Ten-micrometer-thick sections were cut using a Cryostat Microtome System. Tissue sections were fixed with cold acetone for 30 min at −20℃, blocked with 5% BSA, and stained with biotin-PNA (Vector Laboratories), APC-labeled anti-IgD (11–26 c.2a; Thermo Fisher Scientific), and BV510-labeled anti-CD4 (RM4-5; BD Biosciences), followed by FITC-labeled streptavidin (Thermo Fisher Scientific). After each step, the slides were washed at least three times with PBS. Sections were fixed and mounted with an antifade kit (P0123, Beyotime Biotechnology) and then examined using a confocal fluorescence microscope. The images were processed with Imaris and Image J software.

## Detection of PDK1 activity

For detection of PDK1 activity, WT SMARTA mice were i.v. injected with 200 µg of GP61 peptide to prime the SMARTA CD4$^+$ T cells. Sixteen hours later, the splenic were isolated and were first stained with surface markers and then were stimulated with anti-CD3e (2 µg/ml, 145–2 C11; BioXcell), anti-CD28 (1 µg/ml, BE0015, BioXcell), anti-ICOS (2 µg/ml, C398.4A, Biolegend), or anti-CD25 (2 µg/ml, BE0012, BioXcell) at 37℃, 5% $CO_2$ for 1 hr. Stimulated cells were immediately fixed with Phosflow Lyse/Fix buffer (558049, BD Biosciences), incubated with rabbit anti-p-AKT$^{T308}$ (D25E6; Cell Signaling Technology), and sequential stained with fluorochrome-conjugated donkey anti-rabbit IgG (poly4064; BioLegend).

## Proliferation and apoptosis assays

For cell proliferation assay, the mice with indicated genotype were given 2 mg of BrdU i.p. 3 hr before sacrifice. Cells were first stained with surface markers and then were fixed and permeabilized with Cytofix/Cytoperm Fixation/Permeabilization Solution Kit (554714, BD Biosciences). Next, cells were intracellularly stained with anti-BrdU antibody using the BrdU Flow Kit (BD Biosciences) according to manufacturer's introduction. For apoptosis assays, the cells were first stained with surface markers, followed by staining with Caspase-3 (Thermo Fisher Scientific) at 37℃ for 1 hr.

## Transwell migration chemotaxis assay

On 8 dpi, total splenocyte samples from WT and *Pdk1*$^{fl/fl}$::*Cd4*-Cre mice were subjected to depletion of cells that were positive for lineage markers (Lin$^+$ cells) using biotin-conjugated antibodies (anti-CD8 [53–6.7], anti-B220 [RA3-6B2], anti-CD11c [N418], anti-Gr.1 [RB6-8C5], anti-TER119 [TER-119], and anti-NK1.1 [PK136], all from Thermo Fisher Scientific) coupled with the BeaverBeads Mag Streptavidin (Cat # 22305; Beaver). The Lin$^-$ cells were then surface stained with anti-CD4, anti-CD44, and anti-CXCR5 to identify Tfh cells. Next, $4 \times 10^5$ Tfh cells from WT or *Pdk1*$^{fl/fl}$::*Cd4*-Cre mice were loaded into the upper chamber of a 24-well transwell plate (5 µm pore, Corning), and 600 µl of chemotaxis medium supplemented with or without the 1 µg/ml of CXCL13 (583906, Biolegend) was added to the lower chamber. The cells were allowed to migrate for 3 hr in a 5% $CO_2$ incubator at 37℃. Then, all the migrated cells were collected from the lower chamber, and the numbers of migrated Tfh cells were determined by flow cytometry. Based on the absolute number of Tfh cells, the 'net migration (% of input)' was calculated as follows: Net migration (% of input) = (# of migrated Tfh cells to CXCL13 − # of migrated Tfh cells in the absence of CXCL13)/(# of Tfh cells in the input sample).

## RNA-seq and data processing

*Pdk1*$^{fl/fl}$::*Cd4*-Cre and WT mice were infected with LCMV Armstrong, and on 8 dpi, CD4$^+$CD44$^+$-SLAM$^{lo}$ Tfh cells were sorted and total RNA was extracted. Two biological replicates were obtained for both genotypes of mice and used for RNA-seq analysis. Read quality was checked for each sample using FastQC (v0.11.5). Reads were then mapped to the reference genome (mm9) using TopHat (v2.1.1). Raw alignment counts were calculated with R/Bioconductor package GenomicRanges (v1.36.1). Differential expression analysis was performed with DESeq2 (v1.24.0) from the counts. RPKM was calculated, and upregulated or downregulated genes in *Pdk1*$^{fl/fl}$::*Cd4*-Cre Tfh cells were identified by |log$_2$FoldChange|≥0.5 and false discovery rate<0.05.

## Quantitative RT-PCR

Tfh cells were sorted from the spleens of *Pdk1*<sup>fl/fl</sup>::*Cd4*-Cre and WT mice on day 8 post-LCMV Armstrong infection. Total RNA was extracted and reverse-transcribed, and target gene transcripts were measured with quantitative PCR. The primers were listed in **Key Resources Table** above.

## Gene set enrichment analysis

GSEA was performed with GSEA software (version 3.0) with default settings from the Broad Institute and used to determine enrichment of gene sets in WT or *Pdk1*<sup>fl/fl</sup>::*Cd4*-Cre Tfh cells. The gene set of 'Tfh cell differentiation', 'Up-regulated under anti-CD28', 'Down-regulated under anti-CD28', 'Up-regulated under anti-ICOS-L', 'Down-regulated under anti-ICOS-L', 'Raptor-activated genes', 'Raptor-suppressed genes', 'Rictor-activated genes', and 'Rictor-suppressed genes' were collected from previously studies (*Zeng et al., 2016*; *Hao et al., 2018*; *Riley et al., 2002*; *Künzli et al., 2020*).

## Statistical analysis

The statistical significance of differences between groups was determined using two-tailed, unpaired Student's t-test with 95% confidence intervals unless otherwise indicated and GraphPad Prism software (version 8.0). Differences with p-values≥0.05 were considered non-significant (NS). p-values<0.05 were considered significant (*p<0.05; **p<0.01; ***p<0.001).

# Additional information

### Funding

| Funder | Grant reference number | Author |
|---|---|---|
| National Key Research and Development Program of China | 2017YFA0104401 | Shuyang Yu |
| National Natural Science Foundation of China | 31970831 | Shuyang Yu |
| National Natural Science Foundation of China | 31630038 | Shuyang Yu |
| National Natural Science Foundation of China | 31571522 | Shuyang Yu |
| National Natural Science Foundation of China | 31422037 | Shuyang Yu |
| National Natural Science Foundation of China | 81770105 | Weiping Yuan |
| State Key Laboratory of Agro-biotechnology, China Agricultural University | 2019SKLAB6-6 | Shuyang Yu |
| State Key Laboratory of Agro-biotechnology, China Agricultural University | 2019SKLAB6-7 | Shuyang Yu |
| State Key Laboratory of Agro-biotechnology, China Agricultural University | 2018SKLAB6-30 | Shuyang Yu |

The funders had no role in study design, data collection and interpretation, or the decision to submit the work for publication.

### Author contributions

Zhen Sun, Software, Formal analysis, Investigation, Visualization, Methodology, Project administration; Yingpeng Yao, Software, Validation, Investigation, Visualization, Writing - original draft, Writing - review and editing; Menghao You, Software, Investigation, Methodology, Writing - original draft, Writing - review and editing; Jingjing Liu, Formal analysis, Validation, Investigation, Methodology; Wenhui Guo, Formal analysis, Validation, Investigation; Zhihong Qi, Validation, Investigation; Zhao

Wang, Software, Formal analysis, Investigation; Fang Wang, Software, Formal analysis, Validation, Investigation, Methodology; Weiping Yuan, Resources, Formal analysis; Shuyang Yu, Conceptualization, Resources, Supervision, Funding acquisition, Validation, Writing - original draft, Project administration, Writing - review and editing

### Author ORCIDs
Zhen Sun (iD) http://orcid.org/0000-0001-5380-6831
Yingpeng Yao (iD) https://orcid.org/0000-0002-5415-2443
Fang Wang (iD) http://orcid.org/0000-0002-2604-4429
Weiping Yuan (iD) http://orcid.org/0000-0001-8288-5022
Shuyang Yu (iD) https://orcid.org/0000-0002-4686-3296

### Ethics
Animal experimentation: This study was performed in strict accordance with the recommendations in the Guide for the Care and Use of Laboratory Animals of the China Agricultural University. All of the animals were handled according to protocols approved by the Institutional Animal Care and Use Committee at China Agricultural University (Ethical approval Number: AW40101202-3-5).

### Decision letter and Author response
Decision letter https://doi.org/10.7554/eLife.61406.sa1
Author response https://doi.org/10.7554/eLife.61406.sa2

## Additional files

### Supplementary files
• Transparent reporting form

### Data availability
Sequencing data have been deposited in GEO under accession number GSE154976.

The following dataset was generated:

| Author(s) | Year | Dataset title | Dataset URL | Database and Identifier |
|---|---|---|---|---|
| Yu S | 2020 | Regulation of Tfh differentiation by PDK1 | https://www.ncbi.nlm.nih.gov/geo/query/acc.cgi?acc=GSE154976 | NCBI Gene Expression Omnibus, GSE154976 |

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

# Appendix 1

**Appendix 1—key resources table**

| Reagent type (species) or resource | Designation | Source or reference | Identifiers | Additional information |
|---|---|---|---|---|
| Genetic reagent (*M. musculus*) | Mouse: C57BL/6J (CD45.2 and CD45.1) | Jackson Laboratory | RRID: IMSR_JAX:000664 | |
| Genetic reagent (*M. musculus*) | Mouse: B6. Cg-Tg(Cd4-cre) 1Cwi/BfluJ (*Cd4*-Cre) | Jackson Laboratory | RRID: IMSR_JAX:022071 | |
| Genetic reagent (*M. musculus*) | Mouse: B6. Cg-Ndor1Tg (UBC-cre/ERT2)1Ejb/2J (*Rosa26$^{CreER}$*) | Jackson Laboratory | RRID: IMSR_JAX:008085 | |
| Genetic reagent (*M. musculus*) | Mouse: B6. SMARTA | R. Ahmed | Emory University | |
| Genetic reagent (*M. musculus*) | Mouse: B6. *Pdk1$^{fl/fl}$* | W. Yuan | Chinese Academy of Medical Sciences and Peking Union Medical College | |
| Cell line (*H. sapiens*) | HEK293T (human embryonic kidney cells) | ATCC | Cat # CRL-3216, RRID: CVCL_0063 | |
| Biological sample (*M. musculus*) | Primary mouse splenocytes | China Agricultural University | | Freshly isolated from mice |
| Biological sample (*M. musculus*) | Primary mouse bone marrow cells | China Agricultural University | | Freshly isolated from mice |
| Biological sample (*M. musculus*) | Primary mouse serum | China Agricultural University | | Freshly isolated from mice |
| Antibody | Rat monoclonal anti-mouse CD19-PE/Cy7 | Thermo Fisher Scientific | Cat # 25-0193-82, RRID: AB_657663 | FACS (1:100) |
| Antibody | Rat monoclonal anti-mouse CD25-PE | Thermo Fisher Scientific | Cat # 12-0251-83; RRID: AB_465608 | FACS (1:100) |
| Antibody | Rat monoclonal anti-mouse CD4-PE/Cy7 | Thermo Fisher Scientific | Cat # 25-0041-82; RRID: AB_469576 | FACS (1:100) |
| Antibody | Rat monoclonal anti-mouse CD4-APC/eFluor 780 | Thermo Fisher Scientific | Cat # 47-0041-82; RRID: AB_11218896 | FACS (1:100) |
| Antibody | Rat monoclonal anti-mouse CD4-BV510 | BD Biosciences | Cat # 563106; RRID: AB_2687550 | IF (1:100) |
| Antibody | Rat monoclonal anti-mouse/human CD44-FITC | Thermo Fisher Scientific | Cat # 11-0441-82; RRID: AB_465045 | FACS (1:100) |
| Antibody | Rat monoclonal anti-mouse/human CD44-APC | Thermo Fisher Scientific | Cat # 17-0441-83; RRID: AB_469391 | FACS (1:100) |
| Antibody | Rat monoclonal anti-mouse CD45.1-APC | Thermo Fisher Scientific | Cat # 17-0453-82; RRID: AB_469398 | FACS (1:100) |
| Antibody | Mouse monoclonal anti-mouse CD45.1- Percp/Cy5.5 | Thermo Fisher Scientific | Cat # 45-0453-82; RRID: AB_1107003 | FACS (1:100) |
| Antibody | Mouse monoclonal anti-mouse CD45.2- APC | Thermo Fisher Scientific | Cat # 17-0454-82; RRID: AB_469400 | FACS (1:100) |

*Continued on next page*

*Appendix 1—key resources table continued*

| Reagent type (species) or resource | Designation | Source or reference | Identifiers | Additional information |
|---|---|---|---|---|
| Antibody | Mouse monoclonal anti-mouse CD45.2- eFluor 506 | Thermo Fisher Scientific | Cat # 69-0454-82 RRID: AB_2637105 | FACS (1:100) |
| Antibody | Rat monoclonal anti-mouse CD62L- BV510 | BioLegend | Cat # 104441; RRID: AB_2561537 | FACS (1:100) |
| Antibody | Rat monoclonal anti-mouse CD62L- APC | Thermo Fisher Scientific | Cat # 17-0621-83; RRID: AB_469411 | FACS (1:100) |
| Antibody | Rat monoclonal anti-mouse/ human CD45R-FITC | Thermo Fisher Scientific | Cat # 11-0452-86; RRID: AB_465056 | FACS (1:100) |
| Antibody | Rat monoclonal anti-mouse CD45R- PerCP/Cy5.5 | Thermo Fisher Scientific | Cat # 45-0451-82; RRID: AB_1107002 | FACS (1:100) |
| Antibody | Rat monoclonal anti-mouse/ human CD45R- Biotin | Thermo Fisher Scientific | Cat # 13-0452-86; RRID: AB_466451 | FACS (1:100) |
| Antibody | Rat monoclonal anti-mouse IgD- APC | Thermo Fisher Scientific | Cat # 17-5993-82; RRID: AB_10598660 | IF (1:100) |
| Antibody | Rat monoclonal anti-mouse/ human GL7- eFluor 450 | Thermo Fisher Scientific | Cat # 48-5902-82; RRID: AB_10870775 | FACS (1:100) |
| Antibody | Armenian hamster monoclonal anti-mouse PD-1- PE | Thermo Fisher Scientific | Cat # 12-9985-82; RRID: AB_466295 | FACS (1:100) |
| Antibody | Rat monoclonal anti-mouse TCR Vα2-PE | Thermo Fisher Scientific | Cat # 12-5812-82; RRID: AB_465949 | FACS (1:100) |
| Antibody | Rat monoclonal anti-mouse/ rat Foxp3-PerCP/Cy5.5 | Thermo Fisher Scientific | Cat # 45-5773-82 ; RRID: AB_914351 | FACS (1:100) |
| Antibody | Rat monoclonal anti-mouse/ rat Foxp3-APC | Thermo Fisher Scientific | Cat # 17-5773-82 ; RRID: AB_469457 | FACS (1:100) |
| Antibody | Rat monoclonal anti-mouse CD138-PE | BD Biosciences | Cat # 553714; RRID: AB_395000 | FACS (1:100) |
| Antibody | Rat monoclonal anti-mouse CD138-BV421 | BD Biosciences | Cat # 562610; RRID: AB_11153126 | FACS (1:100) |
| Antibody | Armenian hamster monoclonal anti-mouse Fas-PE | BD Biosciences | Cat # 561985; RRID: AB_10895586 | FACS (1:100) |
| Antibody | Mouse monoclonal anti-mouse/human Bcl6-PE | BD Biosciences | Cat # 561522; RRID: AB_10717126 | FACS (1:40) |
| Antibody | Rat monoclonal anti-mouse SLAM-PE | BioLegend | Cat # 115904; RRID: AB_10895586 | FACS (1:100) |
| Antibody | Rat monoclonal anti-mouse SLAM-APC | BioLegend | Cat # 115910; RRID: AB_493460 | FACS (1:100) |
| Antibody | Rat monoclonal anti-mouse/ human ICOS-PE/Cy7 | BioLegend | Cat # 313520; RRID: AB_10643411 | FACS (1:100) |
| Antibody | Rat monoclonal anti-mouse IFN-γ-FITC | Thermo Fisher Scientific | Cat # 11-7311-82; RRID: AB_465412 | FACS (1:100) |
| Antibody | Rat monoclonal anti-mouse IL-17a-PE | BD Biosciences | Cat # 559502; RRID: AB_397256 | FACS (1:100) |
| Antibody | Rat monoclonal anti-mouse IL-4-PE/Cy7 | Thermo Fisher Scientific | Cat # 25-7041-80; RRID: AB_2573519 | FACS (1:100) |
| Antibody | Rabbit monoclonal anti-mouse/human TCF1 | Cell Signaling Technology | Cat # 2203; RRID: AB_2199302 | FACS (1:100) |
| Antibody | Rabbit monoclonal anti-mouse/rat/human PDK1 | Cell Signaling Technology | Cat # 3062; RRID: AB_2236832 | FACS (1:100) |

*Continued on next page*

*Appendix 1—key resources table continued*

| Reagent type (species) or resource | Designation | Source or reference | Identifiers | Additional information |
|---|---|---|---|---|
| Antibody | Rabbit monoclonal anti-mouse/human Hif1a | Cell Signaling Technology | Cat # 36169; RRID: AB_2799095 | FACS (1:100) |
| Antibody | Rabbit monoclonal anti-mouse/rat/human FoxO1 | Cell Signaling Technology | Cat # 2880; RRID: AB_2106495 | FACS (1:100) |
| Antibody | Rabbit monoclonal anti-mouse/rat/human p-AKT$^{T308}$ | Cell Signaling Technology | Cat # 13038; RRID: AB_2629447 | FACS (1:100) |
| Antibody | Rabbit monoclonal anti-mouse/rat/human p-AKT$^{S473}$ | Cell Signaling Technology | Cat # 4060; RRID: AB_2315049 | FACS (1:100) |
| Antibody | Rabbit monoclonal anti-mouse/rat/human p-S6$^{S235/236}$ | Cell Signaling Technology | Cat # 4858; RRID: AB_916156 | FACS (1:100) |
| Antibody | Rabbit polyclone anti-mouse/rat/human p-FoxO1/3a | Cell Signaling Technology | Cat # 9464; RRID: AB_329842 | FACS (1:100) |
| Antibody | Rabbit polyclone anti-mouse/rat/human p-PKCζ/λ | Cell Signaling Technology | Cat # 9378; RRID: AB_2168217 | FACS (1:100) |
| Antibody | Rabbit polyclone anti-mouse/rat/human AKT | Beyotime Biotechnology | Cat # AA326 | FACS (1:100) |
| Antibody | Rabbit monoclonal anti-mouse/human GSK3β | Beyotime Biotechnology | Cat # AF1543 | FACS (1:100) |
| Antibody | Rabbit monoclonal anti-mouse/rat/human p-GSK3β$^{S9}$ | Cell Signaling Technology | Cat # 5558 RRID: AB_10013750 | FACS (1:100) |
| Antibody | Rabbit polyclone anti-mouse/rat/human p-STAT3$^{S727}$ | Cell Signaling Technology | Cat # 9134 RRID: AB_331589 | FACS (1:100) |
| Antibody | Rabbit monoclonal anti-mouse/rat/human STAT3 | Cell Signaling Technology | Cat # 4904 RRID: AB_331269 | FACS (1:100) |
| Antibody | Donkey polyclonal anti-rabbit IgG (minimal x-reactivity)-FITC | BioLegend | Cat # 406403; RRID: AB_893531 | FACS (1:1000) |
| Antibody | Donkey polyclonal anti-rabbit IgG (minimal x-reactivity)-AF647 | BioLegend | Cat # 406414; RRID: AB_2563202 | FACS (1:1000) |
| Transfected construct (*M. musculus*) | MIGR1 (MSCV-IRES-GFP) (plasmid) | This paper | N/A | Retrovirus construct to transfect |
| Transfected construct (*M. musculus*) | MIGR1-*Tcf7* overexpressing (plasmid) | This paper | N/A | Retrovirus construct to transfect |
| Transfected construct (*M. musculus*) | MIGR1-*Bcl6* overexpressing (plasmid) | This paper | N/A | Retrovirus construct to transfect |
| Transfected construct (*M. musculus*) | MIGR1-STAT3-CA (constitutive-active) (plasmid) | This paper | N/A | Retrovirus construct to transfect |
| Transfected construct (*M. musculus*) | MIGR1-*Cxcr5* overexpressing (plasmid) | This paper | N/A | Retrovirus construct to transfect |
| Sequence-based reagent | Tcf7_F | This paper | PCR primers | CCCTTCCTGCGGATATAGAC |
| Sequence-based reagent | Tcf7_R | This paper | PCR primers | GGTACACCAGATCCCAGCAT |

*Continued on next page*

*Appendix 1—key resources table continued*

| Reagent type (species) or resource | Designation | Source or reference | Identifiers | Additional information |
|---|---|---|---|---|
| Sequence-based reagent | Cxcr5_F | This paper | PCR primers | CATGGGCTCCATCACATACA |
| Sequence-based reagent | Cxcr5_R | This paper | PCR primers | GGCATGAATACCGCCTTAAA |
| Sequence-based reagent | Bcl6_F | This paper | PCR primers | AGACGCACAGTGACAAACCA |
| Sequence-based reagent | Bcl6_R | This paper | PCR primers | AGTGTGGGTCTTCAGGTTGG |
| Sequence-based reagent | Icos_F | This paper | PCR primers | TGCCGTGTCTTTGTCTTCTG |
| Sequence-based reagent | Icos_R | This paper | PCR primers | CTTCCCTTGGTCTTGGTGAG |
| Sequence-based reagent | Pdcd1_F | This paper | PCR primers | CTGGTCATTCACTTGGGCTG |
| Sequence-based reagent | Pdcd1_R | This paper | PCR primers | AAACCATTACAGAAGGCGGC |
| Sequence-based reagent | Maf_F | This paper | PCR primers | AGCAGTTGGTGACCATGTCG |
| Sequence-based reagent | Maf_R | This paper | PCR primers | TGGAGATCTCCTGCTTGAGG |
| Sequence-based reagent | Hif1a_F | This paper | PCR primers | CCTTAACCTGTCTGCCACTTTG |
| Sequence-based reagent | Hif1a_R | This paper | PCR primers | TCAGCTGTGGTAATCCACTCTC |
| Sequence-based reagent | Gzmb_F | This paper | PCR primers | CAAAGACCAAACGTGCTTCC |
| Sequence-based reagent | Gzmb_R | This paper | PCR primers | CTCAGCTCTAGGGACGATGG |
| Sequence-based reagent | Id2_F | This paper | PCR primers | GTCCTTGCAGGCATCTGAAT |
| Sequence-based reagent | Id2_R | This paper | PCR primers | TTCAACGTGTTCTCCTGGTG |
| Sequence-based reagent | Prdm1_F | This paper | PCR primers | ACAGAGGCCGAGTTTGAAGAGA |
| Sequence-based reagent | Prdm1_R | This paper | PCR primers | AAGGATGCCTCGGCTTGAA |

*Continued on next page*

*Appendix 1—key resources table continued*

| Reagent type (species) or resource | Designation | Source or reference | Identifiers | Additional information |
|---|---|---|---|---|
| Sequence-based reagent | Gata3_F | This paper | PCR primers | CTTA TCAAGCCCAAGCGAAG |
| Sequence-based reagent | Gata3_R | This paper | PCR primers | CATTAGCGTTCCTCC TCCAG |
| Sequence-based reagent | Stat3_F | This paper | PCR primers | CAATACCATTGACC TGCCGAT |
| Sequence-based reagent | Stat3_R | This paper | PCR primers | GAGCGACTCAAAC TGCCCT |
| Peptide, recombinant protein | KLH | Sigma–Aldrich | Cat# H7017 | |
| Peptide, recombinant protein | $GP_{61-80}$ (GLNGPDIYKGVYQFKSVEFD) | Synthesized by ChinaPeptides | N/A | |
| Peptide, recombinant protein | recombinant murine IL-2 | R and D | Cat # 212–12 | |
| Peptide, recombinant protein | recombinant murine IL-7 | R and D | Cat # 217–17 | |
| Commercial assay or kit | Phosflow Lyse/Fix buffer, 5X | BD Biosciences | Cat # 558049; RRID: AB_2869117 | |
| Commercial assay or kit | Phosflow Perm buffer I | BD Biosciences | Cat # 557885 RRID: AB_2869104 | |
| Commercial assay or kit | Caspase-3 Staining Kit | Thermo Fisher Scientific | Cat # 88-7004-42; RRID: AB_2574939 | |
| Commercial assay or kit | Fixation/Permeabilization Solution Kit | BD Biosciences | Cat # 554714 RRID: AB_2869008 | |
| Commercial assay or kit | Dynabeads M-280 Streptavidin | Thermo Fisher Scientific | Cat # 60210 | |
| Commercial assay or kit | Lipofectamine 2000 Reagent | Thermo Fisher Scientific | Cat # 11668019 | |
| Commercial assay or kit | RNeasy Mini Kit | Qiagen | Cat # 74106 | |
| Commercial assay or kit | FastQuant RT Kit | Tiangen | Cat # KR106-02 | |
| Commercial assay or kit | SuperReal PreMix Plus SYBR Green | Tiangen | Cat # FP205-02 | |
| Chemical compound, drug | Freund's Adjuvant, Complete | Sigma–Aldrich | Cat # F5881 | |
| Chemical compound, drug | Tamoxifen | Sigma–Aldrich | Cat # T5648 | |
| Chemical compound, drug | Corn Oil | Sigma–Aldrich | Cat # C8267 | |

*Continued on next page*

*Appendix 1—key resources table continued*

| Reagent type (species) or resource | Designation | Source or reference | Identifiers | Additional information |
|---|---|---|---|---|
| Chemical compound, drug | PMA | Sigma–Aldrich | Cat # P8139 | |
| Chemical compound, drug | Ionomycin | Sigma–Aldrich | Cat # I0634 | |
| Chemical compound, drug | Polybrene | Sigma–Aldrich | Cat # H9268 | |
| Software, algorithm | Flowjo v10.5 | Treestar | RRID: SCR_008520 | |
| Software, algorithm | Graphpad Prism 8 | Graphpad | RRID: SCR_002798 | |
| Software, algorithm | Adobe Illustrator | Adobe | RRID: SCR_010279 | |
| Software, algorithm | GSEA | http://www.broadinstitute.org/gsea/ | RRID: SCR_003199 | |
| Other | 7-AAD | BD Biosciences | Cat # 559925 RRID: AB_2869266 | |
| Other | PNA-FITC | Vector Laboratories | Cat # FL-1071; RRID: AB_2315097 | FACS (1:500) |
| Other | PNA-Biotin | Vector Laboratories | Cat# BA-0074; RRID: AB_2336190 | IF (1:50) |
| Other | Streptavidin-APC/eFluor 780 | Thermo Fisher Scientific | Cat # 47-4317-82; RRID: AB_10366688 | FACS (1:500) |
| Other | Streptavidin-eFluor 450 | Thermo Fisher Scientific | Cat # 48-4317-82; RRID: AB_10359737 | FACS (1:500) |

