## [Decision Letter]

**Acceptance summary:**

Tfh are extremely important cells in that they are key players in T cell dependent-B cell responses. The role of PDK1 has not been previously analyzed with regards to Tfh differentiation. This thorough study explored the role of PDK1 in Tfh cells and further our understanding of these important cells.

**Decision letter after peer review:**

Thank you for submitting your article "The kinase PDK1 is critical for promoting T follicular helper cell differentiation" for consideration by *eLife*. Your article has been reviewed by three peer reviewers, and the evaluation has been overseen by a Reviewing Editor and Tadatsugu Taniguchi as the Senior Editor. The following individuals involved in review of your submission have agreed to reveal their identity: Carolyn Genevieve King (Reviewer #1); Amanda Poholek (Reviewer #2).

The reviewers have discussed the reviews with one another and the Reviewing Editor has drafted this decision to help you prepare a revised submission.

The study by Sun et al. demonstrates a requirement for CD4 T cell intrinsic expression of PDK1 to generate TFH cells in the context of LCMV infection in both non-competitive and competitive conditions. RNAseq analysis of PDK1-deficient TFH shows a defect in the expression of many genes and pathways known to be essential for TFH cells. These findings are further corroborated by biochemical assays showing that the lack of PDK1 leads to impaired PI3K/Akt signaling, GSK3b phosphorylation and mTORC1 activation. Overexpression of TCF1, a transcription factor required for TFH differentiation, partially rescued such defects. The data presented are well presented, and support the conclusions.

Essential revisions:

Evidence about the timing of the effect, the upstream signals (for instance the role of CD28) need to be provided to strengthen the conclusion that PDK1 is required for Tfh. An additional immunization model will also increase the impact of the finding.

Reviewer #1:

The study by Sun et al. demonstrates a requirement for CD4 T cell intrinsic expression of PDK1 to generate TFH cells. Using a T cell conditional knockout model, the authors show that PDK1 deficiency impairs TFH generation during LCMV condition in both non-competitive and competitive (mixed bone marrow chimera) settings. Comparison of wild type and PDK1-/- TFH cells by RNAseq -/- TFH clearly shows a defect in the expression of many genes and pathways known to be essential for TFH cells; these findings are further corroborated by biochemical assays showing impaired PI3K/Akt signaling, GSK3b phosphorylation and mTORC1 activation in PDK1-/- TFH cells. Overexpression of TCF1, a transcription factor required for TFH differentiation, partially rescues TFH differentiation by PDK1-/- T cells suggesting that TCF1 is induced downstream of PDK1 signaling. These findings are generally in line with what is expected and known about PI3K activity/TFH cells, and this is where the authors have focused much of their attention. The manuscript could be improved by investigating whether PDK1 is activated downstream of CD28, ICOS or both; and whether PDK1 may have any PI3K independent role in CD4 T cells as has been suggested for CD8's.

1) The data in Figure 1D nicely show impaired TFH generation by PDK1-/- T cells both as a proportion of total CD4 T cells (gated on CD44?) as well as the number of CD4 T cells. However, based on the numbers reported it seems that Th1 numbers are also significantly decreased, indicating that PDK1 deficiency leads to a general expansion defect in the CD4 T cell compartment. This is not unexpected given previous published data (1), but might be important to understanding the nature of the Tfh defect – is it something that occurs early or late during infection; i.e. is it a defect in expansion of the compartment, initial acquisition or later maintenance of the Tfh fate?

2) All of the TFH markers shown in Figure 1D with the exception of Bcl6 are nicely reproduced in the bone marrow chimera experiments shown in Figure 2D. Bcl6 in Figure 2D only appears to be impacted on a small proportion of the TFH compartment as opposed to a global defect. Do the authors have an explanation for why this might be the case? Does it have something to do with TFH gating? Related to the first comment, it would also be informative to see if these markers are impacted in the non-TFH compartment to determine if their expression is TFH-specific or not.

3) The RNAseq data shown in Figure 3 is an important resource that might be used to understand the signaling pathways directly impacted by PDK1 deficiency in T cells. This could be done by looking at additional publicly available gene enrichment pathways. For example, are there differences in CD28 versus ICOS dependent pathways in these cells? What about mTORC1 versus mTORC2 – specific to TFH cells as reported by Zeng et al. (2).

4) PDK1 KO T cells have defects in pAkt but also have defects in total expression of Akt. Decreased pAKT-S473 can more or less be explained by decreased total expression of AKT; what is interesting, is that pAKT-S308 is decreased independently of AKT expression. This might point to more impairment of mTORC1 compared to mTORC2. This would be consistent with the very strong impairment of pS6 in Figure 4. Although GSK3b phosphorylation is also decreased in PDK1 KO TFH cells (which according to Yang et al. (3) would be more important for mTORC2 activation) the authors should show total GSK3b expression to aid in the interpretation of this data.

5) PDK1 KO TFH cells are partially rescued by TCF7 overexpression but these data do not directly link PDK1 deficiency to TCF1. In the Discussion, the authors suggest that PDK1/Akt activates mTORC1 to drive TCF1 expression but a previous report suggests that TCF1 is regulated by mTORC2 (3). The authors should try to explain/place their findings in the context of what has been published.

6) Work by Doreen Cantrell has shown that PDK1 can have a PI3K independent effect in CD8 T cells and is required for mTORC1 activation (4). It would be interesting to see if this is also the case for CD4 T cells and could help explain some of the discrepancies in the TFH field – namely work showing that mTORC1 is alternately required for and represses TFH cell fate (2,5); the same is true for HIF activation (6-8) (which the authors demonstrate in Figure 3, is impacted by PDK1 deletion). The authors might be in the position to shed light on this question through a careful analysis of whether PDK1 expression/activity is differentially regulated by various signaling pathways which are important for TFH cells (e.g. TCR? CD28? IL-2R? ICOS? >> mTORC1, mTORC2, HIF….). Importantly, the authors have generated a useful tool – SMARTA PDK1^fl/fl^ERT2-Cre mice – which could be used to assess the impact of early versus late deletion of PDK1 in T cells. This would allow them to address whether PDK1 deficiency impacts early vs. late TFH differentiation (e.g. TCF1 induction or maintenance) which would help them to hone in on which signaling pathways to investigate further.

1) Park, S.-G. et al. The kinase PDK1 integrates T cell antigen receptor and CD28 coreceptor signaling to induce NF-κB and activate T cells. Nature Immunology 10, 158-166 (2009).

2) Zeng, H. et al. mTORC1 and mTORC2 Kinase Signaling and Glucose Metabolism Drive Follicular Helper T Cell Differentiation. Immunity 45, 540-554 (2016).

3) Yang, J. et al. Critical roles of mTOR Complex 1 and 2 for T follicular helper cell differentiation and germinal center responses. e*Life* 5, (2016).

4) Finlay, D. K. et al. PDK1 regulation of mTOR and hypoxia-inducible factor 1 integrate metabolism and migration of CD8^+^ T cells. J Exp Med 209, 2441-53 (2012).

5) Ray, J. P. et al. The Interleukin-2-mTORc1 Kinase Axis Defines the Signaling, Differentiation, and Metabolism of T Helper 1 and Follicular B Helper T Cells. Immunity 43, 690-702 (2015).

6) Zhu, Y. et al. The E3 ligase VHL promotes follicular helper T cell differentiation via glycolytic-epigenetic control. J Exp Med 216, 1664-1681 (2019).

7) Dong, L. et al. HIF1alpha-Dependent Metabolic Signals Control the Differentiation of Follicular Helper T Cells. Cells 8, (2019).

8) Cho, S. H. et al. Hypoxia-inducible factors in CD4(+) T cells promote metabolism, switch cytokine secretion, and T cell help in humoral immunity. Proc Natl Acad Sci U S A 116, 8975-8984 (2019).

Reviewer #2:

In the manuscript by Sun et al., the authors describe a cell intrinsic role for PDK1 in Tfh cell differentiation in the context of LCMV infection. PDK1 is part of the PI3K signaling pathway downstream of TCR and costimulatory molecules such as ICOS, known to promote p-AKT and downstream mTOR activation. Several studies have described key roles for ICOS, PI3K, and mTOR in promoting Tfh differentiation, thus the finding that PDK1 is required for Tfh cells is in line with what is known about this pathway and its requirements for Tfh differentiation.

The data presented are well described, clear and support the conclusions. There are several alterations which would add further clarity to the manuscript before publication.

1) For all flow cytometry figures, it would be helpful in the figure legends to explain what gates were used. For example in Figure 1C and E, figure legends state analysis is from spleens, are the authors pre-gating on CD4^+^ cells? Please add this detail for all flow figures.

2) In Figure 3 for RNAseq, the authors state that Tfh cells were sorted from WT and PDK1-KO mice, but do not explain what the sorting strategy is for gating. This should be clearly explained in the text and figure legends, as according to Figure 1, it is not clear how Tfh's would be isolated as there are very few present.

3) In Figure 3C, GSEA plot, please add the Normalized enrichment score (NES), p value and FDR to the figure.

4) What percent of the SMARTA cells given TCF-1 expressing RV were successfully transduced? Can the authors confirm overexpression of TCF-1?

5) If possible, the outlier dots in all 2D flow plots are a bit small and could be bigger to make it easier to see them.

Reviewer #3:

The authors performed LCMV Armstrong infection model to test the effect of T cell Pdk1 on Tfh cell generation. They found that Pdk1 is necessary for Tfh cell generation during acute viral infection. Further, they performed RNA-Seq and the data suggest that PDK1 deficiency resulted decreased Tfh cell transcriptional programming. I have the following comments:

1) It has been well documented that PDK1 plays a central role in many signal transduction pathways including the activation of Akt and PKC. Through its effects on these kinases, PDK1 is involved in the regulation of a wide variety of processes, including cell proliferation, differentiation and apoptosis. It is appreciated that the authors focus on Tfh cell, however, the general phenotype of the Pdk1 knockout mice need to be mentioned including the effects on T cell homeostasis, initial activation and differentiation of other T cell types (e.g. Is in vivo and/or in vitro differentiation of Th1, Th2 and Th17 cells affected?). The authors need to carefully examine the roles of Pdk1 on T cell homeostasis, proliferation and differentiation.

2) Although the authors concluded that “Pdk1 as a critical regulator for Tfh cell development”, only one LCMV-Arm model is not enough for this conclusion, the authors need to test the effect of Pdk1 on Tfh cells by using at least another model, such as NP-OVA or KLH immunization.

3) For the mechanistic study, it will be important to choose early time points (days 2-3) after LCMV infection, to identify primary events, instead of associated consequences after Pdk1 knockout.

4) The authors concluded that “TCF1 serves as a critical regulator at downstream of PDK1 in promoting Tfh cell differentiation” and did rescue exp in Figure 4E. Can CXCR5 or Bcl6 overexpression also rescue the defective generation of Tfh cells by Pdk1 knockout?

5) PI3K signals are important for lymphocyte trafficking. Do Pdk1-deficient T cells have defect in migratory response toward CXCL13 or CCL21?

6) An important effect of Tfh cell function is the induction of germinal center. Do Pdk1 knockout mice have reduced germinal center phenotypes?

---

## [Author Response]

Essential revisions:Evidence about the timing of the effect, the upstream signals (for instance the role of CD28) need to be provided to strengthen the conclusion that PDK1 is required for Tfh. An additional immunization model will also increase the impact of the finding.

We have carefully considered the reviewers’ comments and performed extensive new experiments to address their concerns during the past several months. As a result, we added new data about the timing of the effect, the upstream signal assay, and an additional immunization model which focused on the major concerns from reviewers and editors. We amended the new data and modified the relative description in revised manuscript. The detailed information was also provided in the response to reviewer’s comments below.

Reviewer #1:The study by Sun et al. demonstrates a requirement for CD4 T cell intrinsic expression of PDK1 to generate TFH cells. Using a T cell conditional knockout model, the authors show that PDK1 deficiency impairs TFH generation during LCMV condition in both non-competitive and competitive (mixed bone marrow chimera) settings. Comparison of wild type and PDK1-/- TFH cells by RNAseq -/- TFH clearly shows a defect in the expression of many genes and pathways known to be essential for TFH cells; these findings are further corroborated by biochemical assays showing impaired PI3K/Akt signaling, GSK3b phosphorylation and mTORC1 activation in PDK1-/- TFH cells. Overexpression of TCF1, a transcription factor required for TFH differentiation, partially rescues TFH differentiation by PDK1-/- T cells suggesting that TCF1 is induced downstream of PDK1 signaling. These findings are generally in line with what is expected and known about PI3K activity/TFH cells, and this is where the authors have focused much of their attention. The manuscript could be improved by investigating whether PDK1 is activated downstream of CD28, ICOS or both; and whether PDK1 may have any PI3K independent role in CD4 T cells as has been suggested for CD8's.

We sincerely thank the reviewer for her professional evaluation and constructive suggestion for improving the quality of our manuscript. To address all concerns from the reviewer, we performed extensive experiments and bioinformatic analysis and reorganized our data sets. We believe the current version is more than improved and thereof hope that the current version can be accepted to publish in *eLife*.

1) The data in Figure 1D nicely show impaired TFH generation by PDK1-/- T cells both as a proportion of total CD4 T cells (gated on CD44?) as well as the number of CD4 T cells. However, based on the numbers reported it seems that Th1 numbers are also significantly decreased, indicating that PDK1 deficiency leads to a general expansion defect in the CD4 T cell compartment. This is not unexpected given previous published data (1), but might be important to understanding the nature of the Tfh defect – is it something that occurs early or late during infection; i.e. is it a defect in expansion of the compartment, initial acquisition or later maintenance of the Tfh fate?

We agree with the reviewer for this concern. To directly exhibit the Th1 numbers, we added the statistical data of both Th1 percentage and numbers in current Figure 1D with the fold change in comparison with Tfh cells. As mentioned by the reviewer, our results reflected that numbers of Th1 cells (2.6-fold) were also significantly decreased, though the frequency of Th1 cells in the absence of PDK1 was elevated compared with WT controls. However, more severe defects were shown in both percentage and numbers of Tfh cells (26.5-fold), which particularly suggested the specificity on differentiation of Tfh population even though a general expansion defect existed in the CD4 T cell compartment.

To further confirm the points suggested by the reviewer about the defect in expansion of the compartment, initial acquisition, or late maintenance of the Tfh fate, we applied the adoptive transfer model by using SMARTA *Pdk1*^fl/fl^*ERT2*-Cre mice to analyze early differentiation (current Figure 4A-E) of Tfh cells. The results indicated that PDK1 plays essential roles in general expansion in the CD4 T cell compartment. In addition, we found PDK1 deficiency led to defects in both initial acquisition and later maintenance (current Figure 4F-I) of the Tfh fate. The detailed information was amended in the relative sections of current manuscript.

2) All of the TFH markers shown in Figure 1D with the exception of Bcl6 are nicely reproduced in the bone marrow chimera experiments shown in Figure 2D. Bcl6 in Figure 2D only appears to be impacted on a small proportion of the TFH compartment as opposed to a global defect. Do the authors have an explanation for why this might be the case? Does it have something to do with TFH gating? Related to the first comment, it would also be informative to see if these markers are impacted in the non-TFH compartment to determine if their expression is TFH-specific or not.

We agree with the reviewer for this concern. We noticed that the distinct degree of reduced Bcl6 expression in original Figure 1D and Figure 2D. After reviewing the data set and repeating the experiments, we found the extremely low degree of Bcl6 expression shown in original Figure 1D did not representatively reflect the difference of mean level from at least three independent experiments in primary mice, though the conclusion on reduced Bcl6 expression in PDK1-deficient Tfh cells is reliable. To appropriately present the data, we replaced the original histogram and statistical data in the new version and the conclusion remains consistent, which exhibited that Bcl6 expression (current Figure 1E) was statistically decreased in PDK1-deficient Tfh cells (CXCR5^+^CD44^+^).

On the other aspect, we changed the strategy as Hao et al., 2018, to show the intrinsic regulation of PDK1 in Tfh cells from BM chimeric mice, which showed the contribution rate of both WT and PDK1-deficient donors. We believe that the new figure panels (current Figure 3A-G) are more direct and convincing for the readership and the conclusion remains consistent. Our new data exhibited that Bcl6 expression was significantly decreased but not extremely low in PDK1-deficient Tfh cells (CXCR5^+^CD44^+^).

To address the concern raised by the reviewer that “if these markers are impacted in the non-TFH compartment to determine if their expression is TFH-specific or not”. We analyzed the PD-1, ICOS, and Bcl6 expression in non-TFH compartments (Th1 and naïve CD4 T cells (CD4^+^CD44^-^CD62L^hi^)) and the results are shown in Author response image 1. Given PD-1 and Bcl6 are typical Tfh signature genes, we found that they generally expressed at a much lower level in Th1 or naïve T cells than those in Tfh cells (Figure 1E and F). The expression of ICOS is altered in naïve T cells, but significant lower in PDK1-deficient Th1 cells than those in WT, which is in accordance with the defects in general activation of T cells due to PDK1 deficiency.

**Author response image 1. sa2fig1:** Analysis of PD-1, ICOS, and Bcl^-^6 expression in naive (A), and Th1 (B) cells.

3) The RNAseq data shown in Figure 3 is an important resource that might be used to understand the signaling pathways directly impacted by PDK1 deficiency in T cells. This could be done by looking at additional publicly available gene enrichment pathways. For example, are there differences in CD28 versus ICOS dependent pathways in these cells? What about mTORC1 versus mTORC2 – specific to TFH cells as reported by Zeng et al. (2).

We thank the reviewer for raising this suggestion. In order to examine the differences of CD28 versus ICOS signaling between WT and PDK1-deficient cells, we extracted data from published studies and performed GSEA analysis (Kunzli et al., 2020; Riley et al., 2002). The results suggested that PDK1-deficient Tfh cells showed lower expression of signatures in the gene set “Up-regulated under anti-CD28” containing up-regulated genes upon anti-CD28 stimulation, while exhibited increased expression of signatures in the gene set “Down-regulated under anti-ICOS-L” containing down-regulated genes upon ICOS-L blocking (Figure 5E). These analyses implied that CD28 or ICOS dependent PDK1 activity may involve in Tfh cell differentiation. To further validate this conclusion, we stimulated CD4^+^ T cells isolated from GP61-primed WT SMARTA mice with different stimuli combinations. We found anti-ICOS only could elicit a higher level of p-AKT^T308^, an indicator of PDK1 activity, while combination of anti-ICOS plus anti-CD3 or/and anti-CD28 had a similar effect with anti-ICOS only. These results further suggested ICOS-dependent PDK1 activity is essential for Tfh cells. We showed these figures in current Figure 5F and G, and the text was amended accordingly.

To examine effects of mTORC1 versus mTORC2 between two genotypes of cells, GSEA was also performed with gene sets obtained from published two studies (Hao et al., 2018; Zeng et al., 2016). We observed both Raptor and Rictor-activated genes were significantly enriched in WT Tfh cells, while Raptor and Rictor-suppressed genes were remarkably enriched in *Pdk1*^fl/fl^Cd4-Cre Tfh cells. These results suggested that PDK1 downstream genes involved in both mTORC1 and mTORC2 dependent pathways play crucial roles in regulating Tfh cell differentiation. These results were shown in Figure 6A and the text was amended accordingly.

4) PDK1 KO T cells have defects in pAkt but also have defects in total expression of Akt. Decreased pAKT-S473 can more or less be explained by decreased total expression of AKT; what is interesting, is that pAKT-S308 is decreased independently of AKT expression. This might point to more impairment of mTORC1 compared to mTORC2. This would be consistent with the very strong impairment of pS6 in Figure 4. Although GSK3b phosphorylation is also decreased in PDK1 KO TFH cells (which according to Yang et al. (3) would be more important for mTORC2 activation) the authors should show total GSK3b expression to aid in the interpretation of this data.

We agreed with the reviewer for this concern. Decreased pAKT-S473 expression was also reported in B cell population due to PDK1 deficiency (Venigalla et al., 2013), but the reason remains to be further investigated. Given the total AKT expression is substantially decreased in PDK1-deficient Tfh cells, we cannot exclude the possibility that impaired AKT expression contributes to the decreased pAKT-T308. Nevertheless, the decreased pAKT-T308 is in accordance with conditional ablation of PDK1 in T cells, which is an upstream kinase of AKT (Toker and Newton, 2000). As presumed by the reviewer, PDK1 deficiency has more profound effect on mTORC1 activity, as indicated by remarkable impairment of p-S6 activity. While our GSEA analysis and experiments also indicated strong correlation between PDK1 and mTORC2.

As requested, we also checked the total GSK3β expression in WT and PDK1-deficient Tfh cells on day 8 post infection. We found GSK3β expression was significantly elevated in PDK1-deficient Tfh cells compared with WT cells. This result is consistent with decreased GSK3β phosphorylation and TCF1 expression in PDK1-deficient Tfh cells, which reported by Yang et al. (Yang et al., 2016). We thank the reviewer again for this suggestion and modified the relative figure and descriptions in revised manuscript.

5) PDK1 KO TFH cells are partially rescued by TCF7 overexpression but these data do not directly link PDK1 deficiency to TCF1. In the Discussion, the authors suggest that PDK1/Akt activates mTORC1 to drive TCF1 expression but a previous report suggests that TCF1 is regulated by mTORC2 (3). The authors should try to explain/place their findings in the context of what has been published.

We thank the reviewer for her suggestion. Yang et al. reported that mTORC2-pAKT may also support TCF-1 activity through the inactivation of GSK3β, an inhibitor of β-catenin and TCF-1 (Yang et al., 2016). Correspondingly, we also found impaired p-AKT^S473^ and p-GSK3β activity in PDK1-deficient cells, indicating a positive link between PDK1/AKT and TCF-1. On the other aspect, Xu et al. reported mTORC1-dependent p-STAT3^S727^ activity is crucial for driving TCF1 expression in T follicular regulatory cells (Xu et al., 2017). Due to severe defects of mTORC1 activity, we also checked the p-STAT3^S727^ activity in PDK1-deficient Tfh cells. We found impaired phosphorylation of STAT3 at S727 in PDK1-deficient Tfh cells (Figure 6C). Furthermore, we found that overexpression of STAT3-CA (a constitutive active form of STAT3) could also partially rectify the defects in PDK1-deficient Tfh cells (Figure 6D and E). These analyses further indicated PDK1mTORC1-p-STAT3-TCF1 is also critical for Tfh cell differentiation. We have supplemented these data and relative discussion based on our observations and published results (Xu et al., 2017; Yang et al., 2016).

6) Work by Doreen Cantrell has shown that PDK1 can have a PI3K independent effect in CD8 T cells and is required for mTORC1 activation (4). It would be interesting to see if this is also the case for CD4 T cells and could help explain some of the discrepancies in the TFH field – namely work showing that mTORC1 is alternately required for and represses TFH cell fate (2,5); the same is true for HIF activation (6-8) (which the authors demonstrate in Figure 3, is impacted by PDK1 deletion). The authors might be in the position to shed light on this question through a careful analysis of whether PDK1 expression/activity is differentially regulated by various signaling pathways which are important for TFH cells (e.g. TCR? CD28? IL-2R? ICOS? >> mTORC1, mTORC2, HIF….). Importantly, the authors have generated a useful tool – SMARTA PDK1^fl/fl^ERT2-Cre mice – which could be used to assess the impact of early versus late deletion of PDK1 in T cells. This would allow them to address whether PDK1 deficiency impacts early vs. late TFH differentiation (e.g. TCF1 induction or maintenance) which would help them to hone in on which signaling pathways to investigate further.1) Park, S.-G. et al. The kinase PDK1 integrates T cell antigen receptor and CD28 coreceptor signaling to induce NF-κB and activate T cells. Nature Immunology 10, 158-166 (2009).2) Zeng, H. et al. mTORC1 and mTORC2 Kinase Signaling and Glucose Metabolism Drive Follicular Helper T Cell Differentiation. Immunity 45, 540-554 (2016).3) Yang, J. et al. Critical roles of mTOR Complex 1 and 2 for T follicular helper cell differentiation and germinal center responses. eLife 5, (2016).4) Finlay, D. K. et al. PDK1 regulation of mTOR and hypoxia-inducible factor 1 integrate metabolism and migration of CD8^+^ T cells. J Exp Med 209, 2441-53 (2012).5) Ray, J. P. et al. The Interleukin-2-mTORc1 Kinase Axis Defines the Signaling, Differentiation, and Metabolism of T Helper 1 and Follicular B Helper T Cells. Immunity 43, 690-702 (2015).6) Zhu, Y. et al. The E3 ligase VHL promotes follicular helper T cell differentiation via glycolytic-epigenetic control. J Exp Med 216, 1664-1681 (2019).7) Dong, L. et al. HIF1alpha-Dependent Metabolic Signals Control the Differentiation of Follicular Helper T Cells. Cells 8, (2019).8) Cho, S. H. et al. Hypoxia-inducible factors in CD4(+) T cells promote metabolism, switch cytokine secretion, and T cell help in humoral immunity. Proc Natl Acad Sci U S A 116, 8975-8984 (2019).

We thank the reviewer for giving us this constructive suggestion. We have noticed that in CTLs, PDK1 but not PI3K or AKT is crucial for mTORC1 activity (Finlay et al., 2012). It is worthy to investigate this in CD4 T cells. A recent paper has pointed out that gain-of-function mutations in the gene encoding the phosphatidylinositol-3-OH kinase catalytic subunit p110δ (PI3Kδ) result in enhanced phosphorylation of AKT and S6 in CD4^+^ T cells relative to that in wildtype CD4^+^ T cells (Preite et al., 2018), which implies possible link between PI3K and mTORC1 in CD4^+^ T cells. However, whether PDK1 has PI3K independent effect in CD4^+^ T cells and is required for mTORC1 activation is still needed to be further investigated in future study. In current study, we mainly focused on the effect of PDK1 ablation on downstream signal and consistent with the results from Cantrell Lab, we also observed decreased mTORC1 activity and reduced Hif1a expression (Figure 6C). To address which signaling is responsible for PDK1 activity, we performed GSEA analysis and in vitro validation. As claimed above (reviewer 1, concern 3), the GSEA results showed CD28 or ICOS dependent PDK1 activity may be involved in Tfh cell differentiation. The in vitro validation assay indicated anti-ICOS only could elicit higher level of p-AKT^T308^, an indicator of PDK1 activity, while combination of anti-ICOS plus anti-CD3 or/and anti-CD28 had a similar effect with antiICOS only. Besides, the CD25 signaling did not affect PDK1 activity in our experimental system. These analyses collectively indicated that ICOS signaling had a profound effect on PDK1 activity to control Tfh cell differentiation, which is essential for initiation of Tfh program (Crotty, 2011).

By using adoptive transfer models with SMARTA *Pdk1*^fl/fl^*ERT2*-Cre or WT donors, we analyzed the impact of deletion of PDK1 on early Tfh cell differentiation upon LCMV infection (Figure 4A, E). We also determined the effect of PDK1 on Tfh cells at late stage via tamoxifen induced deletion model (Figure 4F-I). We found PDK1 deficiency influences Tfh cell differentiation and expansion at both initiation and late maintenance. In addition, by performing GSEA analysis with mTORC1/mTORC2-dependent gene sets, we found PDK1 had a similar gene regulation pattern with both mTORC1 and mTORC2. Our results thus demonstrated ICOS-PDK1mTORC1/mTORC2 signaling is essential for Tfh cell differentiation during the entire process.

Reviewer #2:In the manuscript by Sun et al., the authors describe a cell intrinsic role for PDK1 in Tfh cell differentiation in the context of LCMV infection. PDK1 is part of the PI3K signaling pathway downstream of TCR and costimulatory molecules such as ICOS, known to promote p-AKT and downstream mTOR activation. Several studies have described key roles for ICOS, PI3K, and mTOR in promoting Tfh differentiation, thus the finding that PDK1 is required for Tfh cells is in line with what is known about this pathway and its requirements for Tfh differentiation.The data presented are well described, clear and support the conclusions. There are several alterations which would add further clarity to the manuscript before publication.1) For all flow cytometry figures, it would be helpful in the figure legends to explain what gates were used. For example in Figure 1C and E, figure legends state analysis is from spleens, are the authors pre-gating on CD4^+^ cells? Please add this detail for all flow figures.

We thank the reviewer for this suggestion. The strategies for identification of Th1 cells are CD4^+^CD44^+^CXCR5^-^, Tfh cells are CD4^+^CD44^+^CXCR5^+^, GC Tfh cells are CD4^+^CD44^+^CD62LCXCR5^+^PD-1^hi^ or Bcl^-^6^hi^, respectively. As requested, we supplemented the figure legends in relative parts and modified our statement in the Results section as well. The figure legend of Figure 1C and E were modified as follow: “Flow cytometry analysis of CDC44^+^CXCR5^+^ Tfh cells and CD44^+^CXCR5^-^ Th1 cells gated on total CD4^+^ T cells (top panel), or PD-1^hi^CXCR5^+^ GC Tfh cells (middle panel) and Bcl^-^6^hi^CXCR5^+^ GC Tfh cells (bottom panel) gated on CD44^+^CD62L^-^ CD4^+^ T cells from spleens of WT and *Pdk1*^fl/fl^*Cd4*-Cre mice on 8 *dpi*” and “Expression of PD-1, ICOS, and Bcl6 on Tfh cells (CD4^+^CD44^+^CXCR5^+^) was analyzed by flow cytometry”, respectively.

2) In Figure 3 for RNAseq, the authors state that Tfh cells were sorted from WT and PDK1-KO mice, but do not explain what the sorting strategy is for gating. This should be clearly explained in the text and figure legends, as according to Figure 1, it is not clear how Tfh's would be isolated as there are very few present.

We thank the reviewer for this kind suggestion and apologized for our insufficient description in original manuscript. The sorting strategy is SLAM^-^ subset of CD4^+^CD44^high^CD62L^low^ cells and the gate from WT Tfh cells was directly applied for PDK1-deficient samples. Even though the cells in this population from PDK1-deficient mice are composed of defective Tfh cells, it had the potential of differentiation towards Tfh cell. Therefore, the RNA-seq data in this population would be appropriate to reflect the differentially expressed gene (DEGs) in Tfh cells between WT and PDK1-deficient mice.

3) In Figure 3C, GSEA plot, please add the Normalized enrichment score (NES), p value and FDR to the figure.

We thank the reviewer for this kind suggestion and apologized for our incomplete figure presentation. As requested, we add the NES, p value and FDR to all GSEA plots. Since we supplemented a series of new data, we reorganized our figures in the revised manuscript and the original Figure 3C is the current Figure 5C.

4) What percent of the SMARTA cells given TCF-1 expressing RV were successfully transduced? Can the authors confirm overexpression of TCF-1?

We understand the reviewer’s concern. As requested, we showed the contour plot exhibited the efficiency of TCF1 transduction (GFP^+^) as follows (Author response image 2). Besides, we also sorted GFP^+^ donor cells and confirmed TCF1 overexpression by qPCR (Figure 6—figure supplement 1B). In addition, we also exhibited and confirmed the overexpression of STAT3, CXCR5 and Bcl^-^6 (Figure 6—figure supplement 1B and Author response image 2).

**Author response image 2. sa2fig2:** Analysis of transduction efficiency.

5) If possible, the outlier dots in all 2D flow plots are a bit small and could be bigger to make it easier to see them.

We thank the reviewer for this kind suggestion and apologized for our imperfect figure presentation in initial submission. We guessed that the figure mentioned by the reviewer might be original Figure 2B instead of 2D (a representative flow figure), which reflected the defects in Tfh cell differentiation from PDK1-deficient donors in BM chimeric mice. Considering all the contour figures exhibiting with small size outlets, we respectfully ask the reviewer for understanding that we would like to keep the contour figure with small outlets to ensure consistent format for all the contour figures in entire paper. We changed the strategy to show the defects in Tfh cell differentiation from both WT and PDK1-deficient donors in order to clearly present the data from BM chimeric mice. The new data were shown in current Figure 3B-G and the conclusion remains consistent with original data presentation.

We modified the relative section in revised manuscript, accordingly.

Reviewer #3:The authors performed LCMV Armstrong infection model to test the effect of T cell Pdk1 on Tfh cell generation. They found that Pdk1 is necessary for Tfh cell generation during acute viral infection. Further, they performed RNA-Seq and the data suggest that PDK1 deficiency resulted decreased Tfh cell transcriptional programming. I have the following comments:1) It has been well documented that PDK1 plays a central role in many signal transduction pathways including the activation of Akt and PKC. Through its effects on these kinases, PDK1 is involved in the regulation of a wide variety of processes, including cell proliferation, differentiation and apoptosis. It is appreciated that the authors focus on Tfh cell, however, the general phenotype of the Pdk1 knockout mice need to be mentioned including the effects on T cell homeostasis, initial activation and differentiation of other T cell types (e.g. Is in vivo and/or in vitro differentiation of Th1, Th2 and Th17 cells affected?). The authors need to carefully examine the roles of Pdk1 on T cell homeostasis, proliferation and differentiation.

We thank the reviewer for raising this constructive suggestion. Indeed, PDK1 is involved in the regulation of a wide variety of processes. It has been reported that PDK1 deficiency blocks T cell development at the DN4 stage, and PDK1 is required for CD4^+^ T cell proliferation but not survival of CD4^+^ T cell (Hinton et al., 2004; Park et al., 2009). PDK1 is also critical for Th2 cytokine IL-4 expression (Nirula et al., 2006). Besides, targeting PDK1 with Ox40-Cre resulted in more IL-4 and IL-17a production, indicating PDK1 deficiency leads to disorder of Th2 and Th17 cell differentiation (Yu et al., 2015).

To address the concern raised by reviewer, we also checked Th2 and Th17 cells in KLH-immunized mice model (current Figure 2—figure supplement 1). We found both IL-4 and IL-17a production were elevated in PDK1-deficient CD4^+^ T cells, which were consistent with previous reports. To analyze the effect of PDK1 on Th1 differentiation, we directly analyzed the Th1 subsets in our mice models (current Figure 1C and D; Figure 2A and B). We found PDK1 deficient mice had lower cell numbers of Th1 cells. However, based on the fold-changes in Tfh versus Th1 cells due to PDK1 deficiency, we found that PDK1 deficiency resulted in more severe defects in Tfh cells, implying the specific-role of PDK1 in regulating Tfh differentiation and expansion.

Collectively, PDK1 is not only essential for a general expansion of CD4^+^ T cells, but also has specific role in Tfh cell differentiation.

2) Although the authors concluded that “Pdk1 as a critical regulator for Tfh cell development”, only one LCMV-Arm model is not enough for this conclusion, the authors need to test the effect of Pdk1 on Tfh cells by using at least another model, such as NP-OVA or KLH immunization.

We thank the reviewer for this kind suggestion. As requested, we examined the Tfh cell differentiation in both *Pdk1*^fl/fl^*Cd4*-Cre and WT control mice by using KLH immunization. The results reflected that PDK1-deficient CD4^+^ T cell has substantial defects towards Tfh cell differentiation upon KLH immunization (Figure 2).

3) For the mechanistic study, it will be important to choose early time points (days 2-3) after LCMV infection, to identify primary events, instead of associated consequences after Pdk1 knockout.

We thank the reviewer for this constructive suggestion. As requested, adoptive transfer model was applied to determine the early effect on Tfh cell differentiation (early time point, Day 3). We found the PDK1-deficient CD4^+^ T cell has a general expansion defect upon LCMV challenge, especially, more specifically impaired the differentiation of Tfh cells at early stage (Figure 4A-D). The expansion of early Tfh cells was also impaired due to PDK1 deficiency (Figure 4D). In addition， we found the expression of Tfh regulatory genes *Tcf7*, *Bcl6*, *Cxcr5*, and *Pdcd1* was substantially altered (Figure 4E). These data thus suggested PDK1 has indispensable roles during early Tfh differentiation. We believe these data are important to understand the nature of early Tfh differentiation and the PDK1 function during this process. We amended these data and modified the relative description in revised manuscript.

4) The authors concluded that “TCF1 serves as a critical regulator at downstream of PDK1 in promoting Tfh cell differentiation” and did rescue exp in Figure 4E. Can CXCR5 or Bcl6 overexpression also rescue the defective generation of Tfh cells by Pdk1 knockout?

We thank the reviewer for this constructive suggestion. As requested, we performed the rescue experiments with overexpression of CXCR5 or Bcl6 and we found these two factors can also partially rescue the phenotype of Tfh cell differentiation due to PDK1 deficiency (Figure 6D and E). However, these results are predictable since TCF1 was reported as a key regulator upstream of Bcl6 which is the central regulator for Tfh differentiation (Choi et al., 2015; Wu et al., 2015; Xu et al., 2015). These results collectively suggested that TCF1 is only one of multiple regulators downstream of PDK1 which plays essential roles in Tfh cell differentiation, reflecting PDK1-related gene expression program is complicated during Tfh cell differentiation. Based on these new data, we modified our descriptions in the relative section as well.

5) PI3K signals are important for lymphocyte trafficking. Do Pdk1-deficient T cells have defect in migratory response toward CXCL13 or CCL21?

We thank the reviewer for raising this concern. After we learned relative studies (Hao et al., 2018; Moriyama et al., 2014; Weinstein et al., 2018), we selected CXCL13 (ligand for CXCR5) as a chemokine to detect the migratory response of Tfh cells from both PDK1-deficient and Ctrl mice. The results indicated that PDK1-deficient Tfh cells had significant defects in migratory response toward CXCL13. The results were shown in Figure 1G.

6) An important effect of Tfh cell function is the induction of germinal center. Do Pdk1 knockout mice have reduced germinal center phenotypes?

We agree with the reviewer for this concern. Given the substantial defects in GC B cells and plasma cells were detected via surface staining (current Figure 1H and I), we presumed that PDK1 knockout mice have reduced germinal center. To confirm this point, we performed immunochemical staining in spleen section of mice on Day 8 post infection. The substantial defects in the germinal center formation were exhibited in PDK1 deficient mice (Figure 1J). These data collectively reflect PDK1 deficiency impaired Tfh cell differentiation and resulted in the reduced germinal center phenotypes. We modified the relative part in main text as well.